# TORC2-Gad8-dependent myosin phosphorylation modulates regulation by calcium

**Karen Baker[1], Irene A Gyamfi[1], Gregory I Mashanov[2], Justin E Molloy[2], Michael A Geeves[1], Daniel P Mulvihill[1]\***

[1]School of Biosciences, University of Kent, Canterbury, United Kingdom; [2]The Francis Crick Institute, London, United Kingdom

**Abstract** Cells respond to changes in their environment through signaling networks that modulate cytoskeleton and membrane organization to coordinate cell-cycle progression, polarized cell growth and multicellular development. Here, we define a novel regulatory mechanism by which the motor activity and function of the fission yeast type one myosin, Myo1, is modulated by TORC2-signalling-dependent phosphorylation. Phosphorylation of the conserved serine at position 742 (S742) within the neck region changes both the conformation of the neck region and the interactions between Myo1 and its associating calmodulin light chains. S742 phosphorylation thereby couples the calcium and TOR signaling networks that are involved in the modulation of myosin-1 dynamics to co-ordinate actin polymerization and membrane reorganization at sites of endocytosis and polarised cell growth in response to environmental and cell-cycle cues.
DOI: https://doi.org/10.7554/eLife.51150.001

**\*For correspondence:**
D.P.Mulvihill@kent.ac.uk

**Competing interests:** The authors declare that no competing interests exist.

## Introduction

The actin cytoskeleton underpins cellular organization by maintaining cell shape through the transmission of mechanical signals between the cell periphery and the nucleus, thereby influencing protein expression, protein organization and cellular architecture in response to the needs of the cell. Myosins, which are actin-associated motor-proteins, work in collaboration with an array of actin-binding proteins to facilitate global cytoskeletal reorganization and a plethora of other processes including cell migration, intracellular transport, tension sensing and cell division (*O'Connell et al., 2007*). Each of the many classes of myosin contain three distinct domains: an actin-binding ATPase motor domain that exerts force against actin, a lever arm or neck region that contains light-chain-binding IQ motifs, and a tail region that specifies cargo binding and other molecular interactions.

Although the different classes of myosin perform very different cellular functions, they all operate through the same basic mechanism: the motor domain undergoes cyclical interactions with actin, which are coupled to the breakdown of ATP. Each molecule of ATP that is converted to ADP and inorganic phosphate can generate movement along actin of 5–25 nm and force of up to 5 pN. The regulation of acto-myosin motility is multi-faceted (*Heissler and Sellers, 2016a*), combining regulatory pathways that operate through the actin track (historically called thin-filament regulation) and myosin-linked regulation (historically called thick-filament regulation). This latter control is often mediated by phosphorylation of the heavy chain or light chain(s), or by calcium-regulation of light-chain binding (*Heissler and Sellers, 2016b*). Phosphorylation at the conserved 'TEDS' site motif, which is located within the myosin motor domain of class one myosin, affects acto-myosin interaction (*Bement and Mooseker, 1995*), whereas phosphorylation within the tail region of class five myosin controls cargo binding (*Rogers et al., 1999*). By contrast, phosphorylation of class two myosin light chains and/or heavy chain can change the folded state of the heavy chain, thereby

**eLife digest** The cells of animals, yeast and other eukaryotes all contain a network of filaments known as the actin cytoskeleton that helps to maintain cell shape. When a cell grows, it needs to carefully regulate its actin cytoskeleton to allow the cell to increase in size.

A family of proteins known as the myosins bind to actin filaments and act as motors to undertake many cellular activities, which include reorganising the cytoskeleton and moving cargo around the cell. Myosin proteins contain three distinct sections: a motor domain that exerts force against actin filaments, a long region known as the lever arm and a tail region that binds to cargo and other molecules.

Cells control the activities of myosin proteins in several ways. For example, a protein known as calmodulin binds to the lever arm of myosin to regulate how long and stiff it is, and it can also alter how the motor domain behaves. Furthermore, proteins known as kinases add molecules called phosphate groups to specific parts of the myosin proteins in response to changes in the environment surrounding the cell. Previous studies reported that one of these sites is within a section of the lever arm region that binds to calmodulin. However, it remained unclear whether the action of the protein kinase affects how calmodulin regulates myosin.

Baker et al. used biochemical approaches to study a myosin protein from fission yeast called myosin I. The experiments showed that a protein kinase system called TORC2 added phosphate groups to myosin I affecting how the protein interacted with calmodulin and also alter the shape of the lever arm. This in turn directly affected the ability of the yeast cells to modulate their actin cytoskeleton and grow in response to changes in the nutrients available in the surrounding environment.

Myosin proteins and TORC2 play critical roles in many processes happening in healthy cells and these proteins are often badly regulated in many types of cancer. Therefore, these findings may benefit research into human health and disease in the future.

DOI: https://doi.org/10.7554/eLife.51150.002

altering actin interaction and the ability to form filaments (*Redowicz, 2001*; *Kendrick-Jones et al., 1987*; *Pasapera et al., 2015*). Thus, phosphoregulation of myosin can occur in the head, neck and tail regions, as well as in the light chains, and its impact varies across myosin classes and between paralogues within the same class. The impact of phosphorylation upon the motile function of most myosins remains to be established.

The genome of the fission yeast *Schizosaccharomyces pombe* encodes five myosin heavy chains from classes 1, 2, and 5 (*Win et al., 2002*). The single class one myosin (UniProt Accession: Q9Y7Z8), here termed Myo1, is a 135-kDa protein with a motor domain, a neck region (containing two canonical IQ motifs), and a 49-kDa tail region containing a myosin tail-homology-2 domain (MYTH-2), a membrane-binding pleckstrin homology (PH) domain, an SH3 domain and a carboxyl-terminal acidic region. The acidic region associates with, and activates, the Arp2/3 complex to nucleate actin polymerization (*Lee et al., 2000*). The myosin motor has a conserved TEDS site, which is phosphorylated to modulate the protein's ability to associate with actin (*Attanapola et al., 2009*). Myo1 associates with membranes, primarily at sites of cell growth, where it is required for endocytosis, actin organization and spore formation (*Sirotkin et al., 2005*; *Lee et al., 2000*; *Itadani et al., 2006*).

Calmodulin or calmodulin-like light chains associate with the IQ motifs within the myosin neck to regulate both the length and the stiffness of the lever arm (*Trybus et al., 2007*) and the behavior of the motor domain (*Adamek et al., 2008*). Calmodulins are ubiquitous calcium-binding proteins that associate with and regulate the cellular function of diverse proteins. Calcium associates with up to four EF hand motifs within the calmodulin molecule to instigate a conformational change that modulates the molecule's affinity for IQ motifs (*Crivici and Ikura, 1995*).

*Schizosaccharomyces* pombe has two calmodulin-like proteins, Cam1 and Cam2 (*Takeda and Yamamoto, 1987*; *Itadani et al., 2006*). Cam1 is a typical calmodulin that associates with IQ-domain-containing proteins in a calcium-dependent manner to modulate functions as diverse as endocytosis, spore formation, cell division and spindle pole body integrity (*Takeda and Yamamoto, 1987*; *Moser et al., 1995*; *Moser et al., 1997*; *Itadani et al., 2010*). Although Cam2 shares Cam1's

ability to regulate Myo1, Cam2 differs from Cam1 in two important respects: Cam2 is not essential for viability and is predicted to be insensitive to calcium (*Sammons et al., 2011*; *Itadani et al., 2006*). Furthermore, although cells that lack Cam2 show defects in spore formation following sexual differentiation, they have no significant growth-associated phenotypes during the vegetative growth cycle.

From yeast to man, TOR (Target of Rapamycin) signaling plays a key role in modulating cell growth in response to changes in cell-cycle status and environmental conditions (*Laplante and Sabatini, 2012*; *Hartmuth and Petersen, 2009*). The mTOR kinase forms two distinct protein complexes, TOR complex 1 (TORC1) and TOR complex 2 (TORC2), which are each defined by unique components that are highly conserved across species. TORC1 contains the Regulatory Associated Protein of mTOR (RAPTOR), whereas in TORC2, RAPTOR is replaced with the Rapamycin-Insensitive Companion of mTOR (RICTOR). Both TORC1 and TORC2 complexes control cell migration and F-actin organization (*Liu and Parent, 2011*). TORC2 plays a key role in regulating the actin cytoskeleton in yeasts, *Dictyostelium discoideum* and mammalian cells, modulating actin organization and growth in response to cell-cycle progression and the cellular environment (*Jacinto et al., 2004*; *Baker et al., 2016*; *Lee et al., 2005*).

In *S. pombe*, TORC2 recruits and phosphorylates the fission yeast AGC kinase Gad8 (*Matsuo et al., 2003*), a homologue of human SGK1/2 kinase, to regulate cell proliferation, the switch to bipolar cell growth, cell fusion during mating, and the subsequent meiosis (*Du et al., 2016*). The basic principles of the control of the calcium signaling and phosphorylation signaling pathways are understood, but little is known about the interplay between these parallel modes of regulation.

We have used molecular cell biological, biochemical and single-molecule biophysical techniques to identify and characterize a novel TORC2−Gad8-dependent system that regulates the calcium-dependent switching of the binding of different calmodulin light chains to the neck region of Myo1. We define the contribution that each calmodulin makes to the regulation of this conserved motor protein and describe how they affect the conformation of the Myo1 lever arm. We propose that a concerted mode of regulation involving calcium and phosphorylation controls the motility and function of Myo1 in response to cell-cycle progression.

## Results

### *Schizosaccharomyces* pombe myosin-1 is phosphorylated within the IQ neck domain

Phospho-proteomic studies of the fission yeast *S. pombe* (*Carpy et al., 2014*; *Wilson-Grady et al., 2008*) have revealed a conserved phosphoserine residue that is located within the IQ-motif-containing neck region of class I and V myosins (*Figure 1A*). The location of this AGC family kinase consensus phosphoserine site (*Pearce et al., 2010*) has the potential to impact myosin activity and function by affecting light-chain binding and the conformation of the lever arm. We generated polyclonal antibodies that recognized *S. pombe* myosin-1 when phosphorylated at this conserved serine at position 742 (Myo1[S742]). Myo1[S742] phosphorylation was significantly reduced in cells lacking Ste20 (the fission yeast homolog of the core TORC2 component, RICTOR), and abolished in cells lacking the downstream AGC kinase, Gad8. Thus, Myo1[S742] is phosphorylated in a TORC2–Gad8-kinase-dependent manner (*Figure 1B*).

Within cells, Gad8 kinase activity is reduced through the phosphorylation of a conserved threonine (T6) residue (*Du et al., 2016*; *Hálová et al., 2013*) (*Figure 1C*). A significant reduction of Myo1[S742] phosphorylation was observed in cells expressing phospho-mimetic Gad8.T6D (*Figure 1D*), which has reduced Gad8 kinase activity (*Du et al., 2016*). *Schizosaccharomyces pombe* cells lacking either TORC2 or Gad8 display defects in actin organization, polarized growth regulation and the control of cell-cycle progression (*Petersen and Nurse, 2007*; *Du et al., 2016*). Similarly, replacing Myo1 serine 742 with a phosphorylation-resistant alanine residue in *myo1.S742A* cells blocked the division of cells that were cultured for an extended period in restricted-growth medium (mean length ± SEM (µm): 6.67 ± 0.3 for wildtype cells; 18.50 ± 1.3 for *myo1.S742A* cells (n > 300)) (*Figure 1E*). Therefore, although Gad8 may not directly phosphorylate Myo1[S742], phosphorylation of this residue is dependent upon the TORC2–Gad8 signaling pathway.

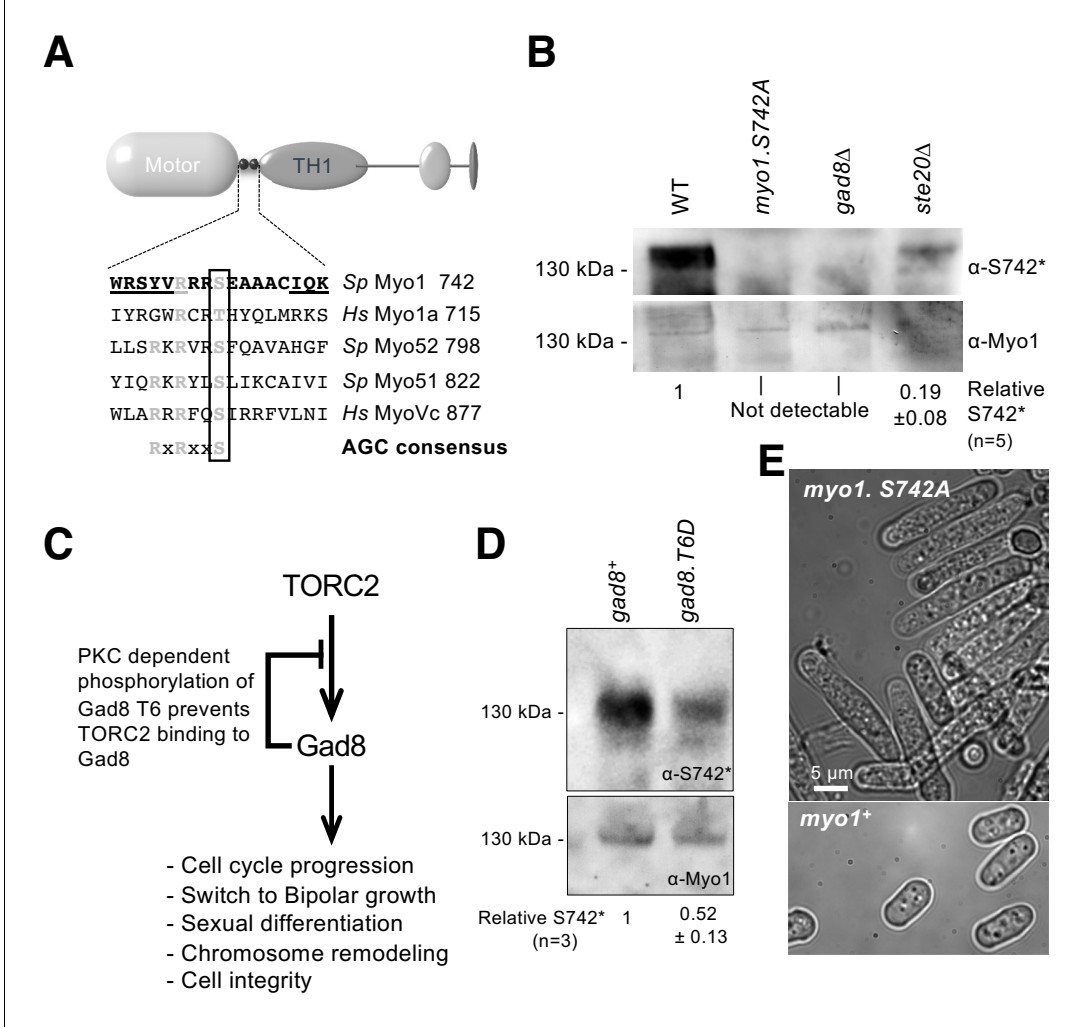

**Figure 1.** Myo1 serine 742 phosphorylation is TORC2 dependent. (**A**) The sequence alignment of myosin IQ regions highlights an AGC kinase consensus sequence that is conserved in class I and V myosins. Underlined residues are those within IQ motifs. (**B**) Western blots of extracts from *myo1+*, *myo1-S742A*, *gad8Δ* and *ste20Δ* cells probed with phospho-specific anti-Myo1$^{S742}$ (upper panel) and anti-Myo1 (lower panel) antibodies demonstrate antigen specificity and a Myo1$^{S742}$ phosphorylation-state dependence upon the TORC2–Gad8 pathway. Ponceau staining was used to monitor equal loading. Relative Myo1$^{S742}$ phosphorylation levels were calculated from five independent equivalent experiments (mean ± sd). (**C**) A schematic of the TORC2–Gad8 signaling pathway. (**D**) Myo1$^{S742}$ phosphorylation is reduced in *gad8.T6D* cells, which have reduced Gad8 kinase activity. Relative Myo1$^{S742}$ phosphorylation levels were calculated from three independent equivalent experiments (mean ± sd). (**E**) Nitrogen-starved wildtype (WT) and *myo1.S742A* cells. In contrast to WT cells, in which growth arrests, *myo1.S742A* cells continue to grow upon nitrogen-starvation-induced G$_1$ arrest. Scale bar: 5 μm.

DOI: https://doi.org/10.7554/eLife.51150.003

We conclude that TORC2-directed Gad8-dependent phosphorylation at S742 regulates Myo1 activity.

## Phosphorylation affects the structure of the lever arm of Myo1

As serine 742 lies within the IQ motif of the Myo1 neck region, we asked whether Myo1$^{S742}$ phosphorylation alters calmodulin binding and the conformation of the neck region. Isoforms of the Ca$^{2+}$-sensitive fission yeast calmodulin (wild type Cam1 and a Cam1.T6C cysteine mutant, allowing conjugation to a fluorescent probe) were purified from bacteria co-expressing the fission yeast NatA amino-α-acetyl-transferase complex in their native amino-terminally (Nt) acetylated forms (*Eastwood et al., 2017*). Two methods were used to measure Ca$^{2+}$-dependent changes in Cam1 conformation. First, a Förster resonance energy transfer (FRET)-based sensor was generated

consisting of N-terminal CyPet donor and C-terminal YPet acceptor fluorophores fused in-frame with Cam1 (*Nguyen and Daugherty, 2005*) (*Figure 2A*). Second, Nt-acetylated Cam1.T6C was conjugated to a cysteine-reactive synthetic fluorophore 2-(4'-(iodoacetamido) anilino naphthalene-6-sulfonic acid (IAANS)). IAANS fluorescence changes in response to changes in its local environment, so the fluorescence emission of this fusion will change in response to calcium-induced changes in Cam1 conformation. The Ca-binding affinity reported by the Cam1-FRET sensor (*Figure 2C*, $pCa_{50}$: 6.12) reflects the global change in Cam1 conformation, whereas the $Ca^{2+}$-dependent change in IAANS' fluorescence signal (*Figure 2C*-inset, $pCa_{50}$: 6.54) reflects changes in the local environment of the amino lobe of Cam1.

Together these probes demonstrated that $Ca^{2+}$ binding induced a change in Cam1 conformation. The rate of $Ca^{2+}$ ion release from Cam1 was independently measured by monitoring changes in the fluorescence of the $Ca^{2+}$ indicator Quin-2 (*Tsien, 1980*). The time-course of $Ca^{2+}$ ion release exhibited three phases, fast, medium and slow, of approximately equal amplitude (rate constants 137, 12.9 and 2.0 $s^{-1}$, respectively), indicating that the cation has different affinities for each $Ca^{2+}$ binding lobe of Cam1 (*Figure 2D*).

To characterize Cam1 binding to the IQ neck region of Myo1, recombinant FRET constructs were produced in which CyPet and YPet were separated by one of the two Myo1 IQ motifs or by both of these motifs (Myo1$^{IQ1}$-FRET, Myo1$^{IQ2}$-FRET or Myo1$^{IQ12}$-FRET) (*Figure 2B* and *Figure 2—figure supplement 1*). Cam1 binding to the IQ motif(s) stabilizes the α-helix and results in a drop in FRET signal in the absence of calcium (*Figure 2E–G*). This drop in signal correlates with a Cam1-bound IQ12 neck region length of 4.6 nm (*Wu and Brand, 1994*), close to the 4.7 nm length predicted from the modeled structure (based upon PDB structure 4R8G). Analysis of interactions between Cam1 and Myo1$^{IQ12}$–FRET revealed two distinct phases to the association of Cam1 molecules with the combined Myo1$^{IQ12}$ motifs. Each phase contributed 50% of the overall change in signal (*Figure 2F*). The first Cam1–Myo1$^{IQ12}$-binding event corresponded to an affinity of less than 0.1 μM (binding was too tight to calculate affinity with higher precision), whereas the second event correlated with an approximately 10-fold weaker binding affinity (0.68 μM). This association was sensitive to calcium (pCa of 5.87) (*Figure 2C*), indicating that Cam1 can only associate with both Myo1 IQ motifs at low cellular $Ca^{2+}$ concentrations. Interestingly, while Cam1 bound tightly to a single, isolated, Myo1$^{IQ1}$ alone ($K_d$ <0.1 μM), no detectable association was observed for the equivalent single Myo1$^{IQ2}$ motif (*Figure 2E*). Together these data are consistent with a sequential cooperative binding mechanism in which the stable residency of Cam1 in the first IQ position is required before calmodulin can bind to Myo1$^{IQ2}$.

Replacing serine 742 within the IQ neck region with a phosphomimetic aspartic acid residue had no significant impact upon the affinity, calcium sensitivity or cooperative nature of the interaction between Myo1 and Cam1 (*Figure 2F*). However, the S742D replacement resulted in a change in maximum FRET signal upon Cam1 binding ($F_{max}$ 46.05 vs 31.64) (*Figure 2F*), indicating that Myo1$^{S742}$ phosphorylation changes the conformation of the lever arm upon Cam1 binding, rather than modulating the affinity of the neck region for Cam1.

## Phosphorylation regulates Myo1 dynamics and endocytosis

Immunofluorescence using Myo1$^{S742}$ phospho-specific antibodies confirmed the presence of serine-742 phosphorylated Myo1 at cortical foci (*Figure 3A*). To explore how this phosphorylation affected Myo1 and calmodulin dynamics in vivo, we generated prototroph *S. pombe* strains in which endogenous *myo1*, *cam1*, or *cam2* genes were fused to cDNA encoding monomeric fluorescent proteins (*Figure 3—figure supplement 1*). Using high-speed (20 fps) single-molecule total internal reflection fluorescence (TIRF) imaging, we explored how Myo1$^{S742}$ phosphorylation impacts Myo1 and Cam1 dynamics and function in vivo. Myo1 and Cam1 associated with the cell membrane in two distinct ways: we observed both (i) rapid, transient, binding of single Myo1 molecules to the cell membrane, characterized by low-intensity, single, stepwise, changes in intensity (*Video 1*), alongside (ii) longer endocytic events that were much brighter and took much longer (*Video 2*).

The rapid, single-molecule, interactions of Myo1 and Cam1 with the membrane had low mobility (0.03 $μm^2.s^{-1}$), ~10 times slower than the diffusion of integral membrane proteins (*Mashanov et al., 2010*). The individual, diffraction-limited fluorescent spots appeared and disappeared at the cell membrane in a single step. The durations of these short single-molecule events (defined as the period over which individual objects were observed and their paths tracked) were exponentially

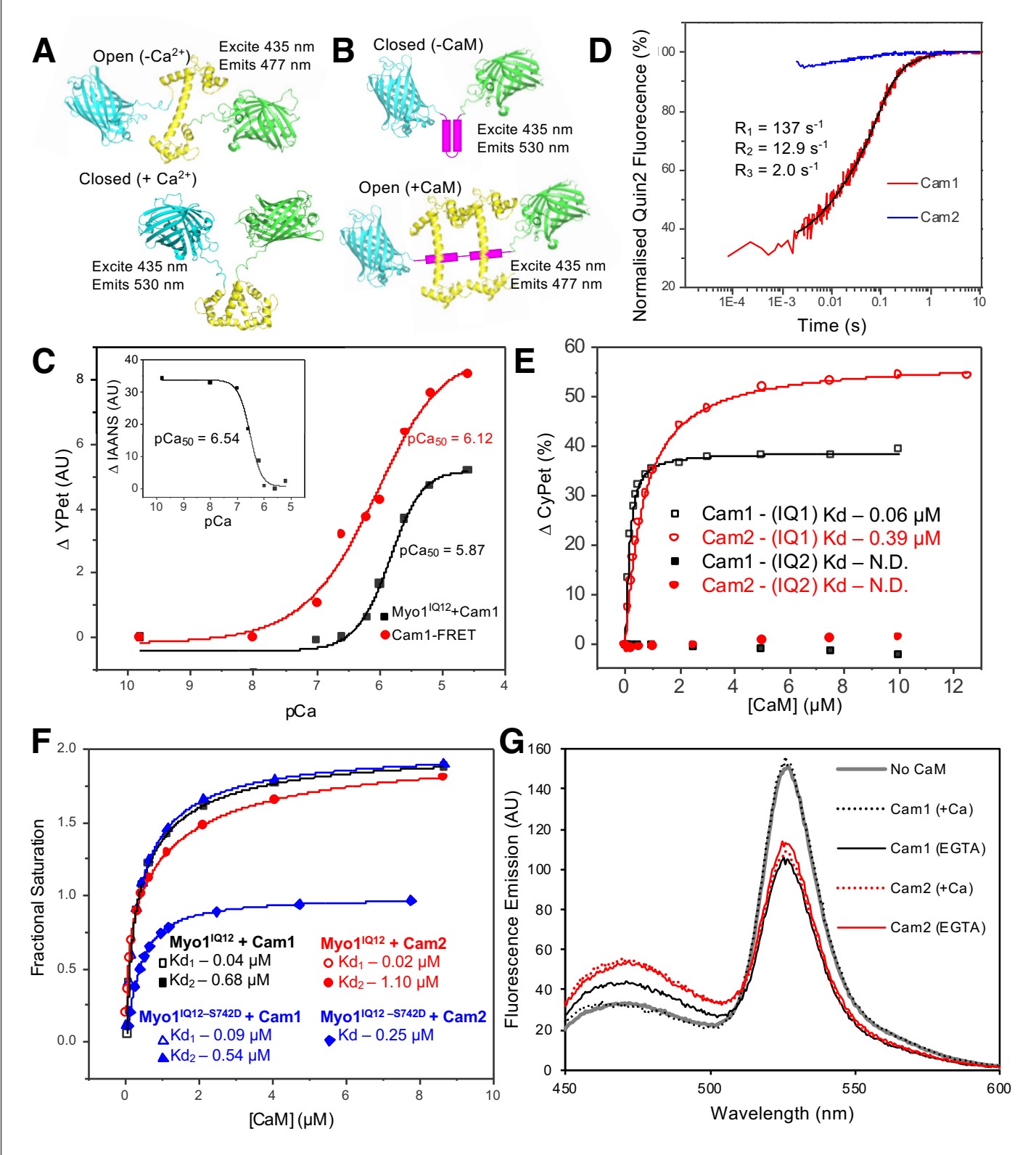

**Figure 2.** In vitro characterization of interactions between Myo1 and Cam1. (**A**) Predicted models of the CyPet–Cam1–YPet FRET reporter protein (Cam1–FRET) in the absence (upper panel) and presence (lower panel) of $Ca^{2+}$. (**B**) Predicted models of the CyPet–Myo1$^{IQ12}$–YPet FRET reporter protein (Myo1IQ12–FRET) in the absence (upper panel) or presence (lower panel) of calmodulin binding (Cypet, cyan; Cam1, yellow; YPet, green; IQ domain, magenta). (**C**) pCa curve plotting $Ca^{2+}$-dependent changes in the acceptor fluorescence (plotted as ΔYPet signal) of the Cam1–FRET protein (red),

*Figure 2 continued on next page*

*Figure 2 continued*

Cam1 association with Myo1$^{IQ12}$-FRET (black) and change in the fluorescence of IAANS-labelled Cam1–T6C (inset). (D) Transient curves of changes in Quin2 fluorescence induced by Ca$^{2+}$ release from Cam1 (red) (with three exponential fit best fit (black)) and from Cam2 (blue) illustrate that only Cam1 associates with Ca$^{2+}$. (E) Curves plotting Cam1- (black) and Cam2-dependent (red) changes of the FRET donor signal of Myo1–FRET proteins containing single IQ domains (IQ1, empty shapes; IQ2, filled shapes) each show that CaM associates with IQ1 but not with an equivalent single IQ2 motif region. (F) Curves plotting Cam1- (squares and triangles) and Cam2-dependent (circles and diamonds) changes in the FRET donor signal of either 0.5 µM wild type (black and red) or S742D phosphomimetic (blue) Myo1$^{IQ12}$–FRET proteins show that although phosphorylation does not significantly impact Cam1 binding, it results in a drop of about 50% in Cam2 interaction. (G) Spectra of 0.5 µM Myo1$^{IQ12}$–FRET reporter alone (grey line) or of 0.5 µM Myo1$^{IQ12}$–FRET reporter mixed with 10 µM saturating concentrations of: Cam1 + Ca$^{2+}$ (black dotted line), Cam1 – Ca$^{2+}$ (black solid line), Cam2 + Ca$^{2+}$ (red dotted line), or Cam1 – Ca$^{2+}$ (red solid line).

DOI: https://doi.org/10.7554/eLife.51150.004

The following figure supplements are available for figure 2:

**Figure supplement 1.** Purified proteins used during in vitro studies.

DOI: https://doi.org/10.7554/eLife.51150.005

**Figure supplement 2.** Cam1 and Cam2 do not interact directly.

DOI: https://doi.org/10.7554/eLife.51150.006

distributed with mean lifetime of ~8 s$^{-1}$ (n = 152) (*Video 1*). The distribution of the durations of individual Myo1 events is shown in *Figure 3—figure supplement 1*. By contrast, during endocytic events, the fluorescence signal increased gradually (at a rate corresponding to ~13 molecules.s$^{-1}$) to a peak amplitude corresponding to ~45 molecules of mNeongreen.Myo1, which persisted for ~6 s (plateau phase), before decaying back to baseline level (at a rate of ~14 molecules.s$^{-1}$) (*Figure 3B*, *Video 2*). The duration (T$_{dur}$) of endocytic events (measured as described in the Materials and methods) was 13.84 s ± 0.39 (mean ± SEM, n = 50) (*Figure 3C*). Although there was significant variation in the maximum mNeongreen.Myo1 intensity (2373 ± 155 AU), there was no correlation between maximum intensity (measured during the plateau phase) and event duration (not shown).

The fluorescence intensity dynamics of Cam1.GFP during endocytic events were similar to those of mNeongreen.Myo1, but T$_{dur}$ was significantly shorter for Cam1.GFP (p<0.0001), 10.99 s +/– 0.21 (n = 52) while the peak (plateau) intensity for Cam1.GFP was roughly double that measured for mNeongreen.Myo1 and equivalent to ~90 GFP molecules (*Figure 3C*), consistent with the occupation of both IQ sites within the Myo1 neck region by Cam1. The briefer event duration observed for Cam1 is best explained by the dissociation of Cam1 from Myo1 before Myo1 leaves the endocytic patch. This process was confirmed by two-color imaging of *mNeongreen.myo1 cam1.mCherry* cells, which revealed how Myo1 and Cam1 arrived simultaneously at the endocytic patch, before Cam1.mCherry disassociated ~3 s before mNeongreen.Myo1 (*Figure 3D*, *Figure 3—figure supplement 1*).

Myo1 and Cam1 dynamics in *myo1.S742A* cells during endocytosis revealed how Myo1$^{S742A}$ had average assembly and disassembly rates and peak intensity measurements that were identical to those of wild-type Myo1, yet the duration of the signal (T$_{dur}$) was 1.5 s shorter for Myo1$^{S742A}$ (12.3 s + /- 0.31 n = 67) (*Figure 3E* and *Figure 3—figure supplement 1*). Consistent with the in vitro data, the *myo1.S742A* mutation did not impact on the ability of Cam1 molecules to bind both IQ motifs, as the average assembly and disassembly rates and the plateau intensity for Cam1 were the same in both wild-type and *myo1.S742A* cells. However, Myo1$^{S742A}$ and Cam1 proteins disassociated simultaneously and somewhat earlier during the endocytic event than in otherwise isogenic wild-type cells.

## Myo1 S742 is phosphorylated in a cell-cycle-dependent manner to regulate polarized cell growth

Upon cell division, fission yeast cells grow exclusively from the old cell end that existed in the parental cell. At a point during interphase (called New End Take Off (NETO)), there is a transition to bipolar growth (*Mitchison and Nurse, 1985*). This cell-cycle switch in growth pattern correlates precisely with a parallel redistribution of endocytic actin patches (*Marks and Hyams, 1985*).

The TIRF imaging data were consistent with widefield, 3D, time-lapse imaging that showed that the lifetimes of Myo1 and Cam1 foci were shorter in *myo1.S742A* cells than in *myo1$^{+}$* cells

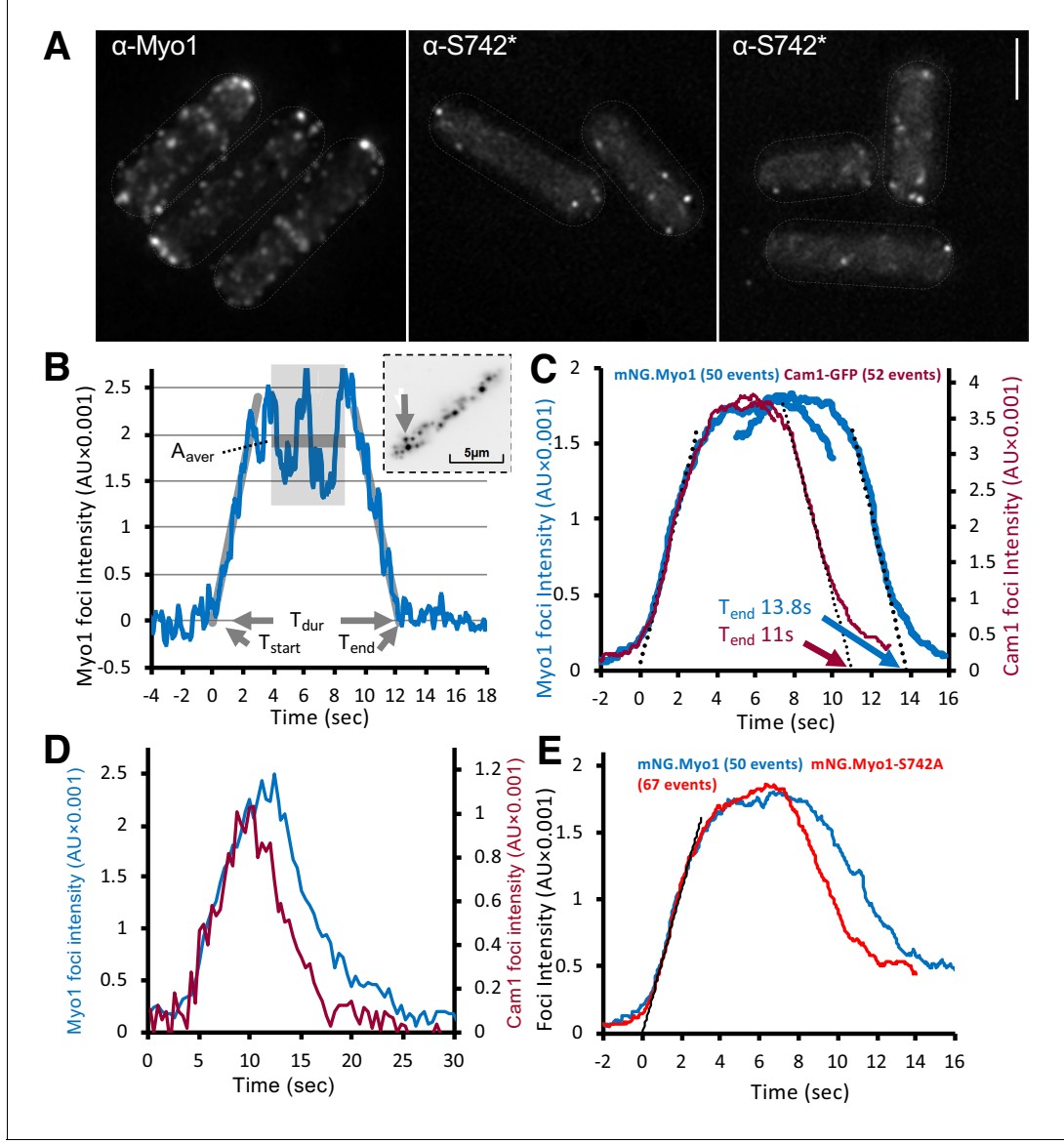

**Figure 3.** Myo1 and Cam1 dynamics in wild-type and *myo1.S742A* cells. (**A**) Maximum projections from 31-z stack widefield immunofluorescence images of wild-type cells, probed with anti-Myo1 (left panel) or anti-Myo1$^{S742*}$ phosphospecific (right panels) antibodies, illustrate that S742 phosphorylated Myo1 localizes to cortical foci (scale bar, 5 μm). (**B**) An example relative-intensity trace of a single mNeongreen.Myo1 endocytic event. Linear fitting (grey lines, 60 points) was used to find the maximum gradient for both the rising and the falling slope. The intercept with zero intensity level was used to calculate $T_{start}$, $T_{end}$, and subsequently, the duration of the event $T_{dur}$. See detailed description in the Materials and methods section. Insert: an arrow highlights the analyzed endocytotic event (5 × 5 pixels area). (**C**) Averaged profile for individual Myo1 (blue) and Cam1 (red) membrane-association events, synchronized relative to $T_{start}$ and $T_{end}$. Dotted lines show fitted rising (Myo1, 537 AU/sec; Cam1, 1073 AU/sec) and falling (Myo1, 567 AU/sec; Cam1, 1028 AU/sec) gradients. (**D**) An example fluorescence trace from simultaneous two-color imaging of a Myo1 (blue line) and Cam1 (red line) membrane-association event observed in *mNeongreen.myo1 cam1.mCherry* cells is consistent with the relative intensities and timings observed using single-fluorophore strains. (**E**) Averaged intensity trajectories of individual Myo1 (blue line) and Myo1.S742A (red line) endocytosis events from TIRFM imaging of *mNeongreen.myo1* and *mNeongreen.myo1.S742A* cells, respectively.
DOI: https://doi.org/10.7554/eLife.51150.007

The following figure supplement is available for figure 3:

**Figure supplement 1.** Relative TIRF profiles.
DOI: https://doi.org/10.7554/eLife.51150.008

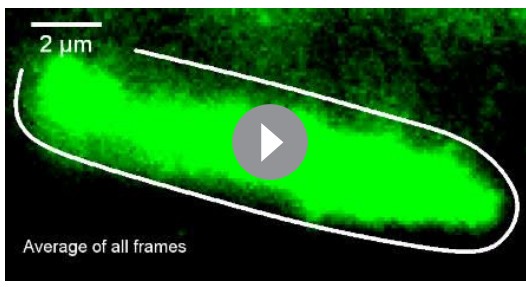

**Video 1.** Single-molecule Myo1 membrane-association events. TIRFM imaging of *mNeongreen.myo1* cell showing rapid single-molecule interactions of Myo1 with the plasma membrane, which are apparent as small bright green spots of a diffraction limited size that are visible over background from the camera noise (highlighted by tracking white lines). These single myosin1 molecules have limited residency time (off rate $7.8 \text{ s}^{-1}$) and mobility ($0.03 \text{ }\mu\text{m}^2.\text{s}^{-1}$) at the plasma membrane. 63 frames per second (fps) @ 23˚C. The plot at the end of the Video was constructed from the analysis of a 50 s full-length video.
DOI: https://doi.org/10.7554/eLife.51150.009

**Video 2.** Endocytic Myo1 events. TIRFM imaging of *mNeongreen.myo1* cells showing endocytosis-associated interactions of Myo1 at the plasma membrane. The myosin1 quickly accumulates at the site of endocytosis, and within 5–8 s of reaching a maximum, rapidly leaves the endocytic site. The accumulated myosin remains immobile on the membrane for the duration of the event. 20 fps @ 23˚C.
DOI: https://doi.org/10.7554/eLife.51150.010

(*Supplementary file 1* Table 1). By contrast, while the *myo1.S742A* allele did not affect the accumulation of Cam2 or LifeACT at sites of endocytosis (*Supplementary file 1* Table 1), the rate of endocytosis (as measured by actin foci lifetimes) differed significantly (p<0.01) between the old and new ends of *myo1-S742A* cells but not of wild-type cells (lifetimes at old end and new cell end: $11.96 \pm 2.28$ and $11.39 \pm 1.07$ for wild-type cells; $14.17 \pm 3.3$ and $11.09 \pm 1.29$ s for *myo1.S742A* cells (mean ± s.d.)). Therefore, although Myo1$^{S742}$ phosphorylation does not impact the assembly of Myo1–Cam1 endocytic foci, it regulates myosin-1 to modulate the activity and function of the ensemble of endocytic proteins during bipolar growth.

As the *myo1.S742A* allele only has affected actin dynamics at the old-cell end during bipolar growth, we examined whether this post-translational modification was subject to cell-cycle-dependent variance. Analysis of extracts from cell-division cycle mutants arrested in $G_1$ (*cdc10. v50* cells) prior to NETO (*Marks et al., 1986*) or in late $G_2$ (*cdc25.22* cells) after NETO revealed that Myo1$^{S742}$ is phosphorylated in a cell-cycle-dependent manner (*Figure 4A*). This was confirmed by monitoring Myo1$^{S742}$ phosphorylation in cells that were synchronized with respect to cell-cycle progression (*Figure 4B-D*). These data established that, at its peak in early interphase (prior to the transition to a bipolar growth pattern), approximately half of cellular Myo1 is phosphorylated on S742, before dropping to undetectable levels by the end of late $G_2$ (the Cdc25 execution point), prior to entry into mitosis. *myo1.S742A* cells have a longer average length than wild-type cells ( $9.77 \pm 1.77$ μm for *myo1$^+$*; $13.2 \pm 2.47$ μm for *myo1. S742A*; t-test >99% significance; n > 500).

In addition to the NETO phenotype, a significant proportion of *myo1.S742A* cells exhibited significant issues with their ability to maintain, linear, polarized, growth, as 24.7% of these cells developed a bent morphology (i.e. growth deviates by >5˚ from the longitudinal axis) (*Figure 4E-F*). The *myo1.S742A* allele did not have an additive effect on the growth-polarity defects associated with cells lacking Tea4, a polarity determinant protein that plays an important role in integrating actin cytoskeleton function with the regulation of polarised cell growth (*Martin et al., 2005*; *Tatebe et al., 2005*) (*Figure 5A*). Consistently, cell-wall staining revealed a significantly higher than normal proportion of *myo1.S742A* cells that exhibited monopolar growth (when compared to equivalent wild type), indicating a disruption in the switch from monopolar to bipolar growth (*Figure 4E-F*). This was confirmed by tracking the cellular distribution of the actin-patch marker, Sla2/End4, following cell division. Sla2 failed to redistribute to the newly divided end of *myo1.S742A* cells during interphase (*Figure 5B*).

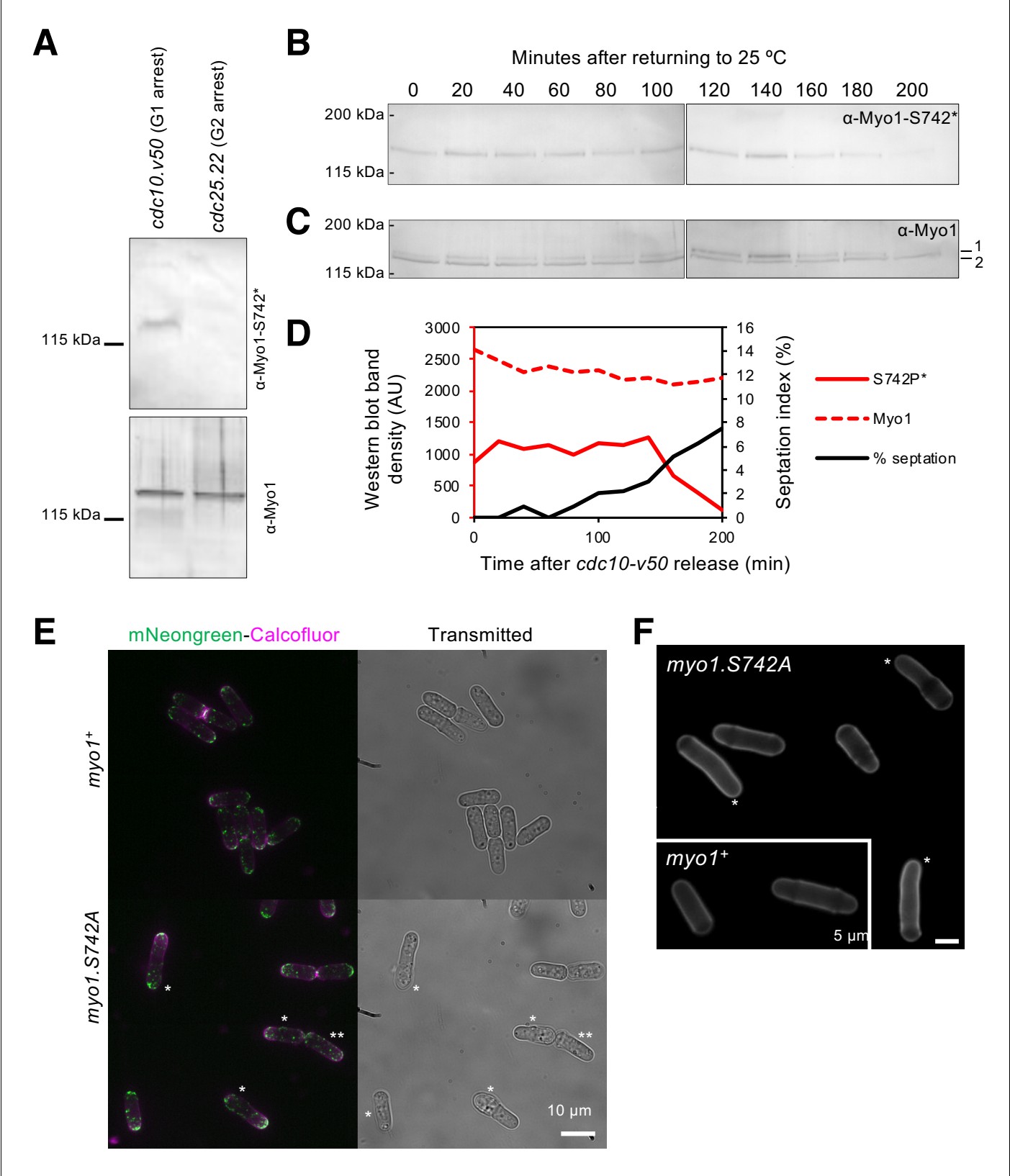

**Figure 4.** Myo1 S742 is phosphorylated in a cell-cycle-dependent manner to affect polarized growth. (**A**) Western blots of extracts from $G_1$-arrested *cdc10.v50* cells and pre-mitotic $G_2$-arrested *cdc25.22* cells probed with phospho-specific anti-Myo1$^{S742}$ (upper panel) and anti-Myo1 (lower panel) antibodies demonstrate that Myo1 $^{S742}$ phosphorylation occurs before the Cdc10 execution point in monopolar $G_1$ cells, and is not detectable by the

*Figure 4 continued on next page*

*Figure 4 continued*

Cdc25 execution point at the end of $G_2$ (n = 3). (B–D) A *cdc10.v50* culture was synchronized in $G_1$ by shifting to 36°C for 240 min before returning to 25°C at time 0. Samples of cells were taken every 20 min from the release and processed for western blotting to monitor Myo1[S742] phosphorylation. The membrane was first probed with phosphospecific anti-Myo1[S742]* antibodies (B), and subsequently probed with anti-Myo1 antibodies to monitor total Myo1 (C). Both phosphorylated (1) and non-phosphorylated (2) Myo1 bands can be observed in panel (C). Equal loading was monitored by Ponceau staining of the membrane. (D) Densitometry measurements of phosphorylated Myo1[S742] (from panel (B)) and total Myo1 (both bands from panel (C)) are plotted along with the % of cells in the culture with septa. (E) Myosin-1 distribution (green), calcofluor-stained regions of cell growth (magenta), and cell outline (transmitted image) of prototroph *mNeongreen.myo1+* and *mNeongreen.myo1.S742A* cells cultured in EMMG medium at 34°C. Asterisks highlight cells that have morphology defects. Scale bar, 10 µm. (F) Calcofluor-stained WT and *myo1.S742A* cells. Asterisks highlight long bent cells displaying monopolar growth. Scale bar, 5 µm.

DOI: https://doi.org/10.7554/eLife.51150.011

This failure of *myo1.S742A* cells to switch to bipolar growth, a and restrict growth upon nutrient depletion (*Figure 1E*) is consistent with the reduced growth rate at the end of the log phase and with growth to an overall higher density upon reaching the stationary phase (*Figure 5C*).

We conclude that cell-cycle-dependent changes in Myo1[S742] phosphorylation modulate the ability of the myosin lever arm region to regulate endocytosis and polarized growth.

## Cam2 associates with internalized endosomes and not with Myo1 during vegetative growth

Myo1 has been reported to associate with a second calmodulin-like protein, Cam2, via its second IQ motif (*Sammons et al., 2011*). However, our data indicate that Cam1 occupies both Myo1 IQ motifs during endocytosis. Widefield microscopy revealed that Myo1 and Cam1 dynamics (*Figure 6A*) at endocytic foci differ significantly from Cam2 dynamics at these foci. Cam2 is recruited to sites of endocytosis later than Myo1 and Cam1, but prior to vesicle scission/budding, whereupon, like CAP-ZA[Acp1], Sla2 and actin, Cam2 remains associated with laterally oscillating, internalized, endosomes (*Figure 6B–C*). Similarly, simultaneous imaging of Cam1 and Cam2 in *cam1.mCherry cam2.gfp* cells revealed how each protein localizes to a significant proportion of foci lacking the other calmodulin, thereby highlighting the different timings of the engagement of each molecule with the endocytic machinery (*Figure 6D*). Finally, although Cam1 recruitment to endocytic foci is abolished when Myo1 is absent (*Figure 6E*), the intensity, volume and number of Cam2 foci actually increases in the absence of Myo1 (*Figure 6F Supplementary file 1* Table 1), even though the internalization and lateral 'oscillating' dynamics of Cam2 and actin were dependent on Myo1 (*Figure 6F & G*). We assume that this arises from the requirement for prior action of Cam1 for vesicle budding.

TIRF imaging revealed that, on average, a total of ~30 Cam2 molecules were recruited to each endocytic focus (compared to 45 and 90 molecules observed for Myo1 and Cam1, respectively), and that the kinetics of Cam2 recruitment to foci differed significantly to those observed for both Myo1 and Cam1. The Cam2 signal often increased steadily, before an abrupt decline (*Figure 7A*), which contrasts with the more gradual (sigmoidal) rise and decay in intensity observed for Myo1 and Cam1 (*Figure 3C, E*). TIRF microscopy (TIRFM) confirmed that Cam2 continued to be associated with the endocytic vesicles after they were internalized and their connection with the cell membrane was broken (*Video 3*). Background-corrected intensity traces of Cam2 dynamics at the membrane patch before, during, and after the end of endocytosis showed that the signal rapidly dropped to baseline (<1 s) (*Figure 7A*), with the Cam2-labelled vesicles remaining visible close to the membrane but moved inwardly, away from the location of the endocytic event. A large number of these mobile, internalized Cam2-labelled vesicles were seen moving within the cytoplasm with relatively low cytosolic background signal (*Video 3*), indicating that much of the Cam2 was associated with endocytic vesicles and remained bound to mature endosomes. We conclude that endocytosis was inhibited, with Cam2 persisting on the endosome while Myo1 remained at the plasma membrane during and after endosome abscission, as previously reported (*Figure 6A*, *Video 2*) (*Sirotkin et al., 2010*; *Berro and Pollard, 2014*; *Picco et al., 2015*). Thus, although Cam1 and Cam2 both localize to sites of endocytosis, they appear to do so at different times, and each have different Myo1 dependencies.

To correlate Myo1-Cam1 dynamics at sites of endocytosis with the internalization of the mature endosome into the cytoplasm, we followed Cam1 and Cam2 dynamics simultaneously in *cam1.*

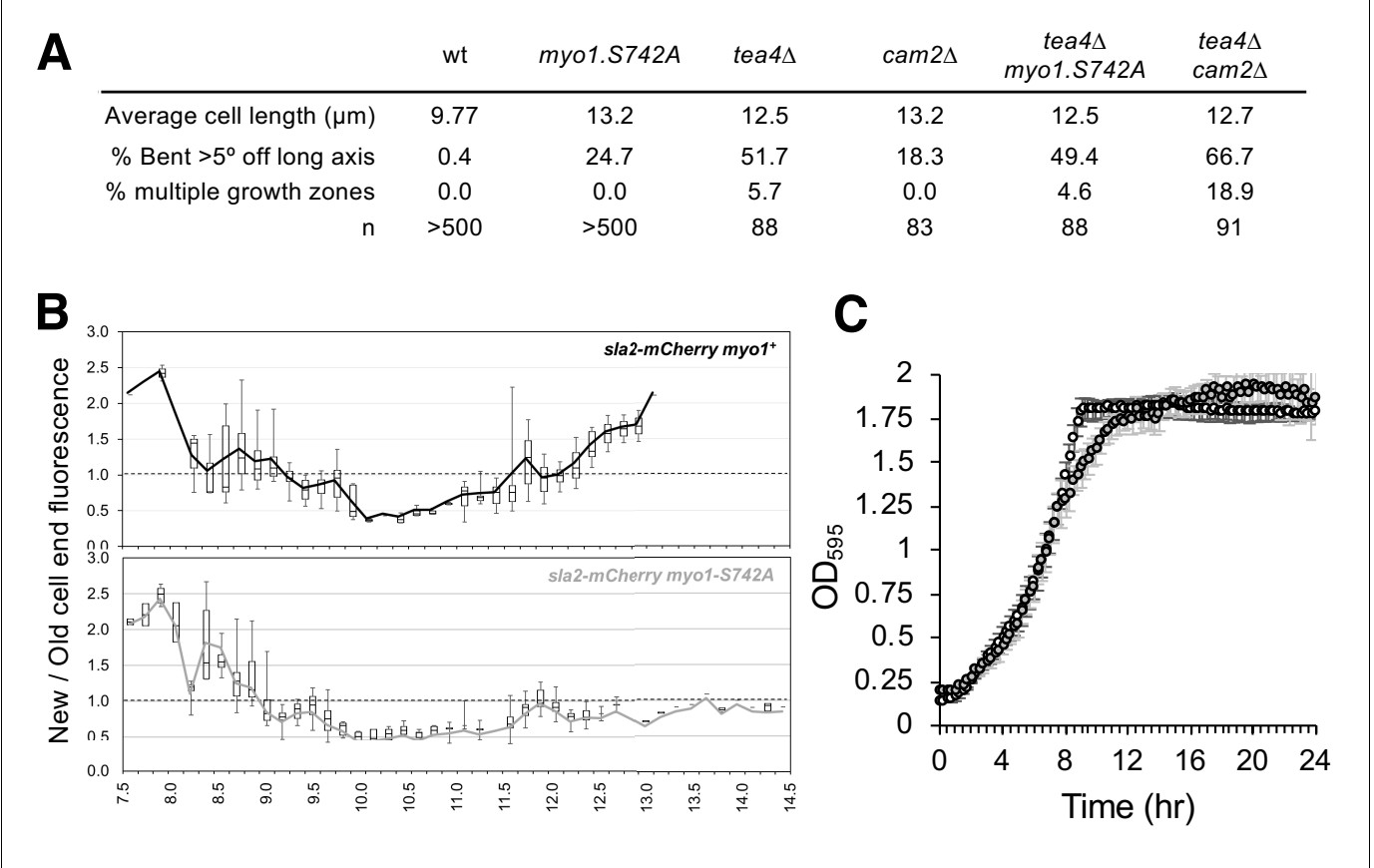

**Figure 5.** Myo1[S742] phosphorylation impacts polarized cell growth. (**A**) Average length and frequency of growth defects in WT, *myo1.S742A*, *tea4Δ*, *cam2Δ*, *tea4Δ myo1.S742A*, and *tea4Δ cam2Δ* cells. (**B**) Ratio of Sla2-mCherry fluorescence at 'new':'old' cell ends, averaged from >30 growing mid-log *sla2-mCherry myo1⁺* (upper panel) and *sla2-mCherry myo1.S742A* (lower panel) cells. Boxes plots show the median and quartile for each length measured, whereas the lines are plots of the mean values at each length measured. (**C**) Averaged growth curves from three independent experiments of prototroph wild-type (empty circles) and *myo1.S742A* (gray-filled circles) cells cultured in EMMG medium at 34°C. Slower growth is apparent at the end of log phase in *myo1.S742A* cells, which grow more until reaching the stationary phase. Error bars denote the s.d. of the mean.
DOI: https://doi.org/10.7554/eLife.51150.012

*mCherry cam2.gfp* cells (*Video 4*). An average curve (*Figure 7B*), generated from profiles of 65 two-color individual endocytic events, synchronized relative to the $T_{start}$ of Cam1 (see *Figure 3B, C*), shows that Cam2 moves away from the cell surface shortly after Cam1 leaves but before Myo1 leaves, with the time of abscission ($T_{scis}$) occurring on average 11.4 s after the event starts ($T_{start}$). Therefore endosome scission takes place immediately prior to the Myo1 disassembly phase (*Figure 3B*), and around the time when Cam1 dissociates from Myo1 (*Figure 3C*). Intriguingly, although the overall distribution of Myo1 and Cam1 appeared to be unaffected in *cam2Δ* cells, the number, volume and intensity of foci were significantly reduced (*Figure 7C, D*; *Supplementary file 1* Table 1).

## Serine 742 phosphorylation increases the affinity of a single Cam2 molecule for Myo1

In vitro analysis revealed how two Cam2 molecules can associate with the unphosphorylated Myo1[IQ12] region (*Figure 2F*) in a process that has two distinct phases. In contrast to Cam1, in which the two Myo1[IQ12] binding events contributed equally to the change in FRET signal, for Cam2, 70% of the signal change was brought about by a single binding event, associated with an affinity of 1.10 μM. The smaller amplitude and tighter binding signal is not accurately measurable, but the combined change in signal is consistent with two binding events.

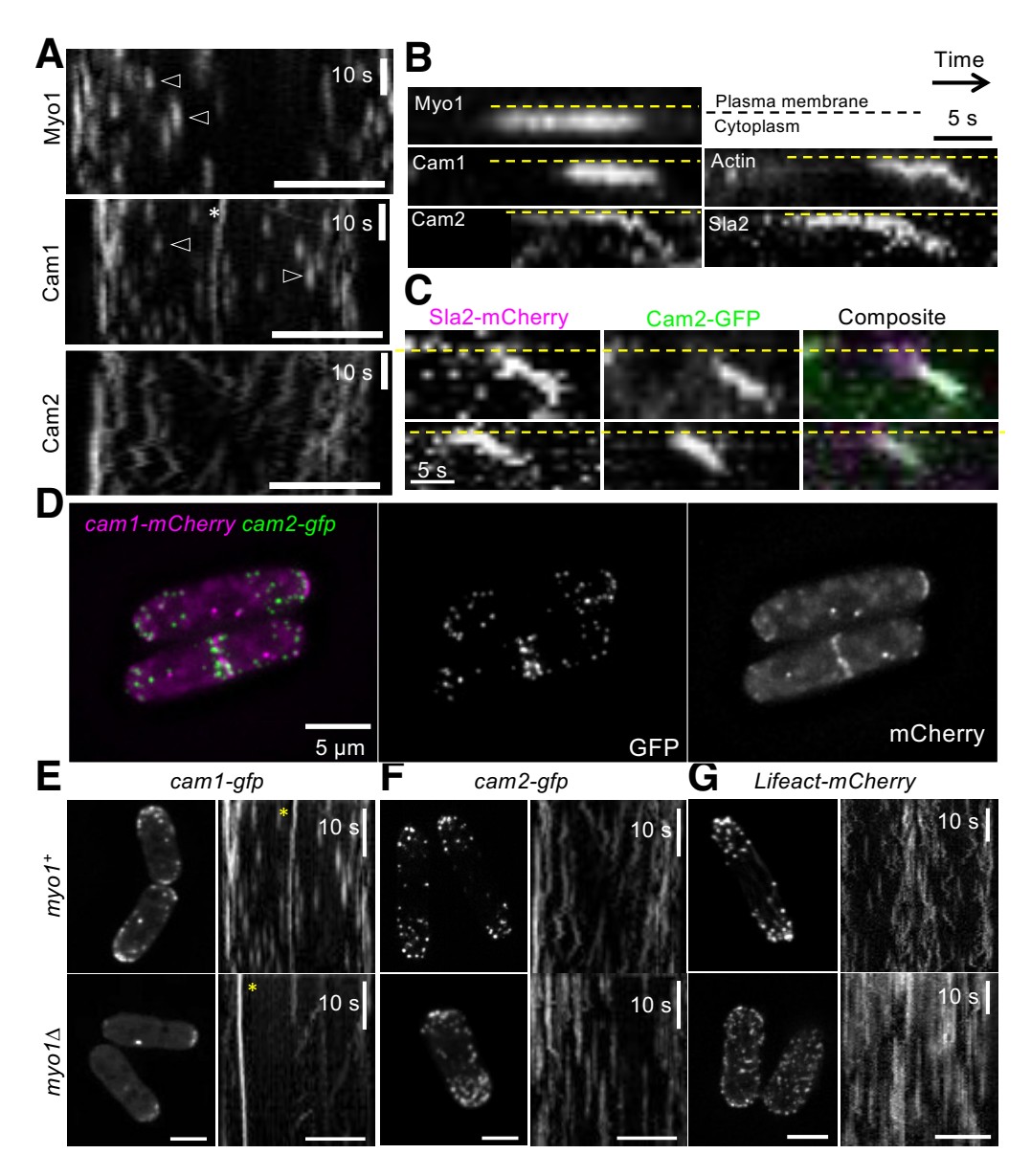

**Figure 6.** Cam2 associates with internalized endocytic vesicles. (A) Kymographs of GFP-labelled foci from maximum projections of 13-z plane time-lapse images of *mNeongreen.myo1* (upper panel), *cam1.gfp* (middle panel) and *cam2.gfp* (bottom panel) cells illustrate the static nature of Myo1 and Cam1 endocytic foci when associated with the plasma membrane (arrowheads). Cam1 foci that are associated with a spindle pole body (SPB) are highlighted. By contrast, Cam2 foci displayed extensive lateral movements. (B) Kymographs generated from single z-plane time-lapse images of single endocytic foci surfaces during vesicle formation and subsequent internalization. Myo1 and Cam1 only associate with the plasma membrane, whereas Cam2, Sla2 and actin are internalized on the vesicle after scission. These kymographs are not aligned temporally. (C) Kymographs of Cam2 and Sla2 co-internalization in *sla2.mCherry cam2.gfp* cells. (D) Maximum projection of a 31-z slice image of *cam1.mCherry cam2.gfp* cells reveals that Cam1 (magenta) and Cam2 (green) colocalize in a subset of endocytic foci. (E–G) Single frames (left panels) and kymographs (right panels) from maximum projections of 13-z plane time-lapse images of *cam1.gfp* (E), *cam2.gfp* (F) and *LifeACT.mCherry* (G) in either *myo1⁺* (upper panels) or *myo1Δ* (lower panels) cells. These images show that although only Cam1 recruitment to endocytic foci is dependent upon Myo1, the myosin is required for the internalization of Cam2-GFP and LifeACT.mCherry foci. Scale bar, 5 μm.

DOI: https://doi.org/10.7554/eLife.51150.013

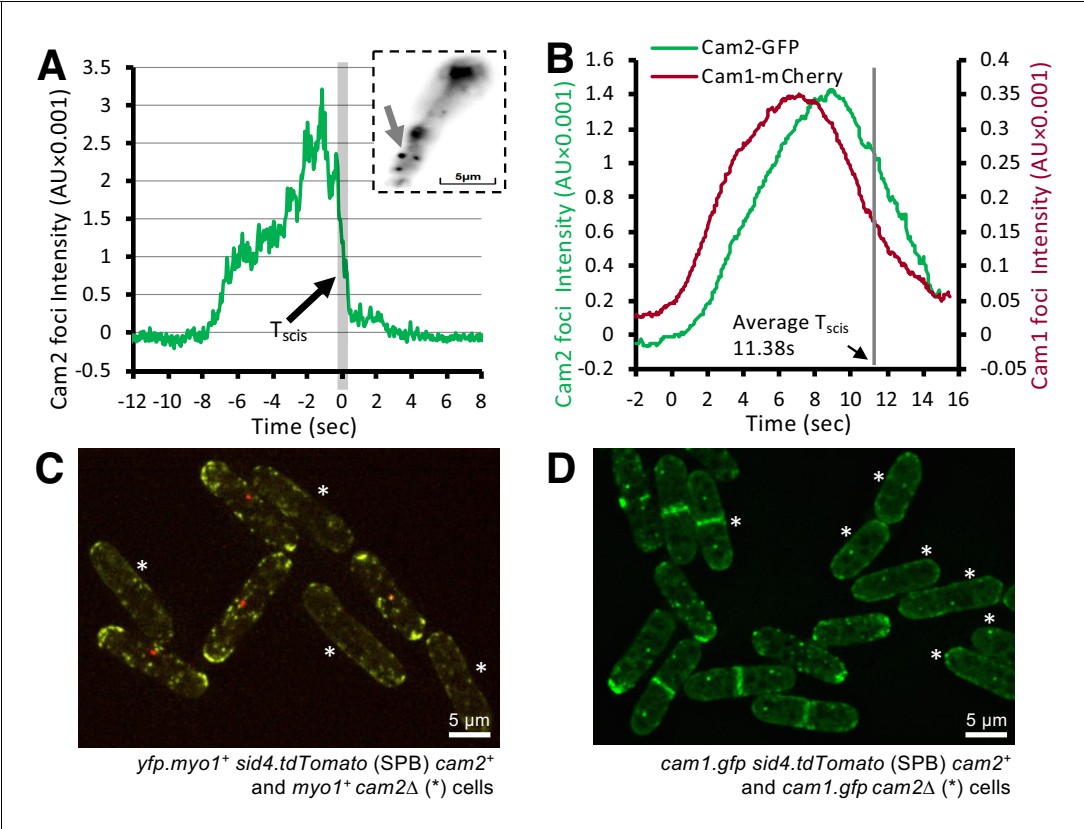

**Figure 7.** Cam2 does not impact Myo1 or Cam1 dynamics in vegetative cells. (A) An example fluorescence trace of a Cam2 membrane binding and vesicle internalization event from TIRFM imaging of *cam2.gfp* cells. An abrupt drop in the fluorescence was marked as 'scission time' ($T_{scis}$, gray vertical line). Insert: an arrow shows the location of the monitored endocytic event ($5 \times 5$ pixels area). (B) Averaged profile from 65 individual Cam2 membrane association events (green line), together with the averaged Cam1-mCherry profile (red) from two-color TIRFM imaging of *cam1.mCherry cam2.gfp* cells. The events were synchronized relative to the Cam1 $T_{start}$. The gray line denotes the mean time of vesicle scission ($T_{scis}$). See detailed description in the Materials and methods section. (C) Maximum projection of a 31-z slice widefield image of a mixture of *yfp.myo1 sid4.tdTomato* (WT, with a red labeled SPB marker) and *yfp.myo1 cam2Δ* (asterisks) cells. Red-labeled SPBs allow differentiation between *cam2+* and *cam2Δ* cells in the same field. (D) Maximum projection of a 31-z slice widefield image of a mixture of prototroph *cam1.gfp sid4.tdTomato* (WT, with a red-labeled SPB marker) and *cam1.gfp cam2Δ* cells (asterisks). Red-labeled SPBs allow differentiation between *cam2+* and *cam2Δ* cells in the same field. Scale bars, 5 μm.

DOI: https://doi.org/10.7554/eLife.51150.014

As predicted from sequence analysis, Cam2 failed to associate with calcium (*Figure 2D*), and its conformation and interactions with Myo1 were insensitive to the divalent cation (*Figure 2G*). Like Cam1, Cam2 had a higher affinity for the first IQ motif (0.4 μM) than for both IQ1 and IQ2 together, and failed to bind IQ2 alone (*Figure 2E*). Cam1 calcium binding, as measured by IAANS labeling or by change in Quin-2 fluorescence were unaffected by Cam2, whereas gel filtration and fluorescence binding assays provided no evidence of a direct physical interaction between the two proteins (*Figure 2— figure supplement 2*). Interestingly, a difference in fluorescence amplitudes between Cam1 and Cam2 binding to the IQ12 motif indicated an impact upon the conformation of the lever arm

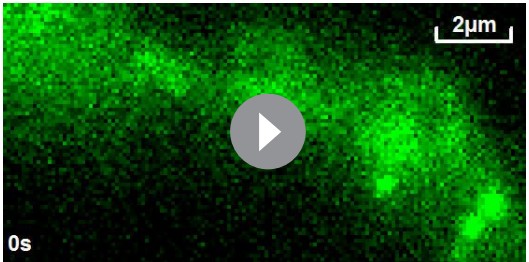

**Video 3.** Endocytic Cam2 events. TIRFM imaging of a *cam2.gfp* cell showing Cam2 recruitment to endocytic vesicles, to which it remains associated after scission and internalization of the endosome. At the start of each event, each spot is immobile, but at the end of the endocytic event, the vesicle oscillates as it is internalized into the cytoplasm. 20 fps @ 23°C.
DOI: https://doi.org/10.7554/eLife.51150.015

(*Figure 2G*), providing a potential mechanism to control Myo1 motor activity directly. However, Myo1$^{S742}$ phosphorylation had no measurable impact upon the dynamics and distribution of Cam2 within *S. pombe* cells undergoing normal vegetative growth (*Figure 8A; Supplementary file 1* Table 1).

## Cam1 and Cam2 associate with Myo1 during meiosis

Calcium levels within log phase yeast cells are relatively low (100–200 nM) (*Ma et al., 2011*; *Miseta et al., 1999*), and so provide conditions that favor the association of Cam1 with Myo1 (pCa, 5.87). Analysis of cell fluorescence indicated the relative abundance of Myo1: Cam1: Cam2 within the *S. pombe* cell to be 0.25: 1.56: 1 (*Supplementary file 1* Table 1), which is similar to the ratio defined by quantitative proteomic analysis of 0.45: 1.56: 1 (*Marguerat et al., 2012*). Similarly, image analysis of Cam1–GFP fluorescence showed how 1.7% of Cam1 associated with discrete foci within cells (*Supplementary file 1* Table 1), 40% of which was dependent upon Myo1, with the majority associating with the SPB (*Figure 6E*). This indicates that ~0.68% of cellular Cam1 associates with Myo1 at dynamic endocytic foci. These relative protein levels, binding affinities and low $Ca^{2+}$ concentrations favor Cam1 binding to Myo1, over Cam2 at both IQ sites (*Figure 8B*), consistent with in vivo observations.

Although $Ca^{2+}$ levels are low during vegetative growth, sporadic prolonged calcium bursts occur upon pheromone release during mating (*Carbó et al., 2017*; *Iida et al., 1990*), and levels elevate significantly (~10 fold) during the subsequent meiosis and sporulation (*Suizu et al., 1995*). Cam1 would be less likely to bind to Myo1 in these conditions (pCa, 5.87). We observed that Myo1$^{S742}$ is phosphorylated in mating and meiotic cells (*Figure 8C*). Cam2 abundance simultaneously increases significantly in relation to Cam1 upon starvation, mating and entry into meiosis (*Mata and Bähler, 2006*; *Mata et al., 2002*). These conditions favor interactions between Myo1 and Cam2 over an association of Cam1 with Myo1 (*Figure 8B*), which is consistent with important roles for both Myo1 and Cam2 at the leading edge of forespore membrane formation during meiosis (*Toya et al., 2001*; *Itadani et al., 2006*). Consistent with this prediction, the lifetimes and dynamics of Myo1, Cam1 and Cam2 foci differ significantly from those observed in vegetative cells (p<0.0001), with foci lasting significantly longer (>1 min) in meiotic and sporulating cells (*Supplementary file 1* Table 1). In contrast to vegetative cells, in cells undergoing meiosis and subsequent spore formation, cortical foci containing accumulations of Cam2 and actin (like those containing like Myo1 and Cam1) were less dynamic, lacking any oscillation and remaining in a fixed position, and had a significantly longer lifetime than foci within actively growing cells (*Figure 8D*, *Supplementary file 1* Table 1, *Videos 5–8*). Consistent with this, endocytosis is significantly diminished in fusing and meiotic cells (*Figure 8E*) containing the stable Myo1, Cam1, Cam2 or actin foci, indicating that the reduction in foci dynamics minimizes endocytosis in meiotic cells.

Finally, we used the *myo1.S742A* allele to monitor the impact of Myo1$^{S742}$ phosphorylation on Myo1, Cam1 and Cam2 dynamics during meiosis. In contrast to wild-type cells, the lifetime of Myo1 and Cam1 foci were not significantly different to each other in *myo1.S742A* cells. In addition, the lifetime of the Myo1 and Cam1 foci in *myo1.S742A* cells were significantly reduced when compared to those in the wild type. Cam2 dynamics did not correlate with Myo1 in *myo1.S742A* cells, which is in contrast to those in the wild type (*Supplementary file 1* Table 1). Myo1 and Cam1 foci were also seen to be shorter in *cam2Δ* cells during meiosis, when compared to those in wild type cells (*Supplementary file 1* Table 1). These data indicate that Myo1$^{S742}$ phosphorylation is required for Cam2 to interact with the Myo1 IQ motif and thereby reduce Myo1 foci dynamics.

The majority of Cam2 foci in meiotic cells lacking Myo1$^{S742}$ phosphorylation remained present in the cell for longer than two mins. Such timing differs significantly from the dynamics of non-phosphorylatable Myo1$^{S742A}$, indicating that normal Cam1 and Cam2 interactions with Myo1 were abolished in Myo1$^{S742A}$ cells. Consistent with observations of *myo1.S742A* cells grown to stationary phase (*Figure 1E*), heterothallic (h$^{90}$) nitrogen-starved $G_1$-arrested *myo1.S742A* cells failed to inhibit polar growth (*Figure 8F*). 27.9% of mating *myo1.S742A* cells continued to grow at their mating (schmooing) tips (vs of *myo1$^+$* 1.8% cells; n > 100), and meioses frequently produced asci with an abnormal number of unequally sized spores (*Figure 8F* , arrowheads) ( 0.9% of *myo1$^+$* asci, 13.1% of *myo1.S742A* asci; n > 100). This spore defect phenotype is reminiscent of the meiotic phenotype of *cam2Δ* cells (*Itadani et al., 2006*), which supports the view that increases in cellular $Ca^{2+}$ and Myo1$^{S742}$ phosphorylation are both key for Cam2 association with and regulation of Myo1.

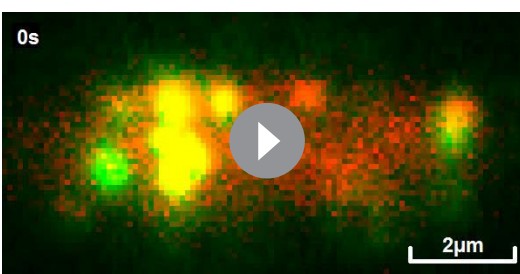

**Video 4.** Comparison of Cam1 and Cam2 dynamics. TIRFM imaging of a *cam1.mCherry cam2.gfp* cell showing early recruitment of Cam1 (red) and subsequent recruitment of Cam2 (green) to the sites of endocytosis. Cam1 disassociates prior to vesicle scission, while Cam2 remains associated with the internalized endosome, so that at the beginning of the endocytic event, the spots are red. During the event, they become yellow as Cam2 (green) is recruited to the vesicle. At the end of the event, the vesicle becomes green as Cam1 dissociates from the endocytotic site. 20 fps @ 23°C.
DOI: https://doi.org/10.7554/eLife.51150.016

These data support a model in which changes in calcium levels and TORC2-dependent phosphorylation status provides a simple two-stage mechanism for modulating motor activity. In this mechanism, modification of lever arm conformation and a switch in calmodulin light-chain preference co-ordinate myosin function with the changing environmental and cell-cycle-dependent needs of the cell (*Figure 8B, F*).

## Discussion

Myosins are subject to diverse modes of regulation, including modulation of the composition of the actin track, changes to cargo and light-chain interactions, and phosphorylation that changes the core physical properties of the motor. Here, we describe a newly discovered mechanism through which phosphorylation of the myosin heavy chain (*Figure 1*) regulates light chain specificity, and lever arm conformation and flexibility, to impact upon cellular function. During the vegetative life cycle, at basal levels of cellular calcium, *S. pombe* Myo1 preferentially associates with two molecules of the calcium-regulated calmodulin light-chain Cam1 (*Figures 2* and *3*). During early stages of the cell cycle, phosphorylation of the Myo1 neck region (*Figure 8G*) changes the conformation of the Cam1-associated lever arm to moderate motor activity, thereby regulating the rate of endocytosis, and a switch from monopolar to bipolar growth (*Figure 5*).

There is a significant increase in TORC2 and Gad8 activity upon starvation, which promotes the onset of the meiotic lifecycle (*Hálová et al., 2013*; *Laboucarié et al., 2017*; *Martín et al., 2017*). Upon starvation, there is an increase in Myo1 serine 472 phosphorylation (*Figure 8*), and *myo1. S742A* cells fail to arrest growth in response to starvation (*Figures 1E* and *Figure 8*F). Phosphorylation of the IQ region, combined with an increase in cytosolic $Ca^{2+}$ levels observed during $G_1$, starvation and meiosis, switches light chain preference to favor the recruitment of a single molecule of the calcium-insensitive calmodulin-like Cam2. However, it is worth noting that there are currently differences of opinion on the relationship between levels of Gad8 activity and cytosolic calcium (*Cohen et al., 2014*). The structures of the IQ region of Myosin-1 and calmodulin (*Lu et al., 2015*) suggest that phosphorylation of Myo1[S742] is likely to impact Cam2 binding at the 1st IQ position. Furthermore, our data reveal that CaM is unable to associate with IQ2 alone, as occupancy of IQ1 is required before a second CaM can bind to IQ2 (i.e. regulatory cooperative binding occurs). This switch in light-chain occupancy may provide a mechanism to change the stiffness of the Myo1 neck region (i.e. the 'lever arm') and might thereby modulate the movement and force that is produced by this region during the acto-myosin ATPase cycle and/or the load-sensitivity of its actin-bound form.

Observations within budding yeast indicate that motor activity from a ring of myosins at the lip of the endosome (*Mund et al., 2018*) is necessary for endocytic internalization, but the mechanism by which the myosin interacts with actin to facilitate this localized activation is unknown (*Sun et al., 2006*). Here we find that the size of the early endocytic patch determines the number of Myo1 molecules necessary to generate a critical local concentration of Arp2/3-nucleated actin filaments (*Barker et al., 2007*). At the critical concentration, myosin heads are able to interact with actin filaments nucleated from either adjacent Myo1 tails or WASP-activated Arp2/3 complexes, which are tethered to the membrane via molecules such as the Talin-like Sla2 (*Sirotkin et al., 2005*; *Sirotkin et al., 2010*). The Myo1 is then primed to act as a tension sensor against the actin filament as it pushes against the membrane of the internalized endosome and grows against the significant 0.85 MPa (8.3 atm) turgor pressure within the cell (*Minc et al., 2009*).

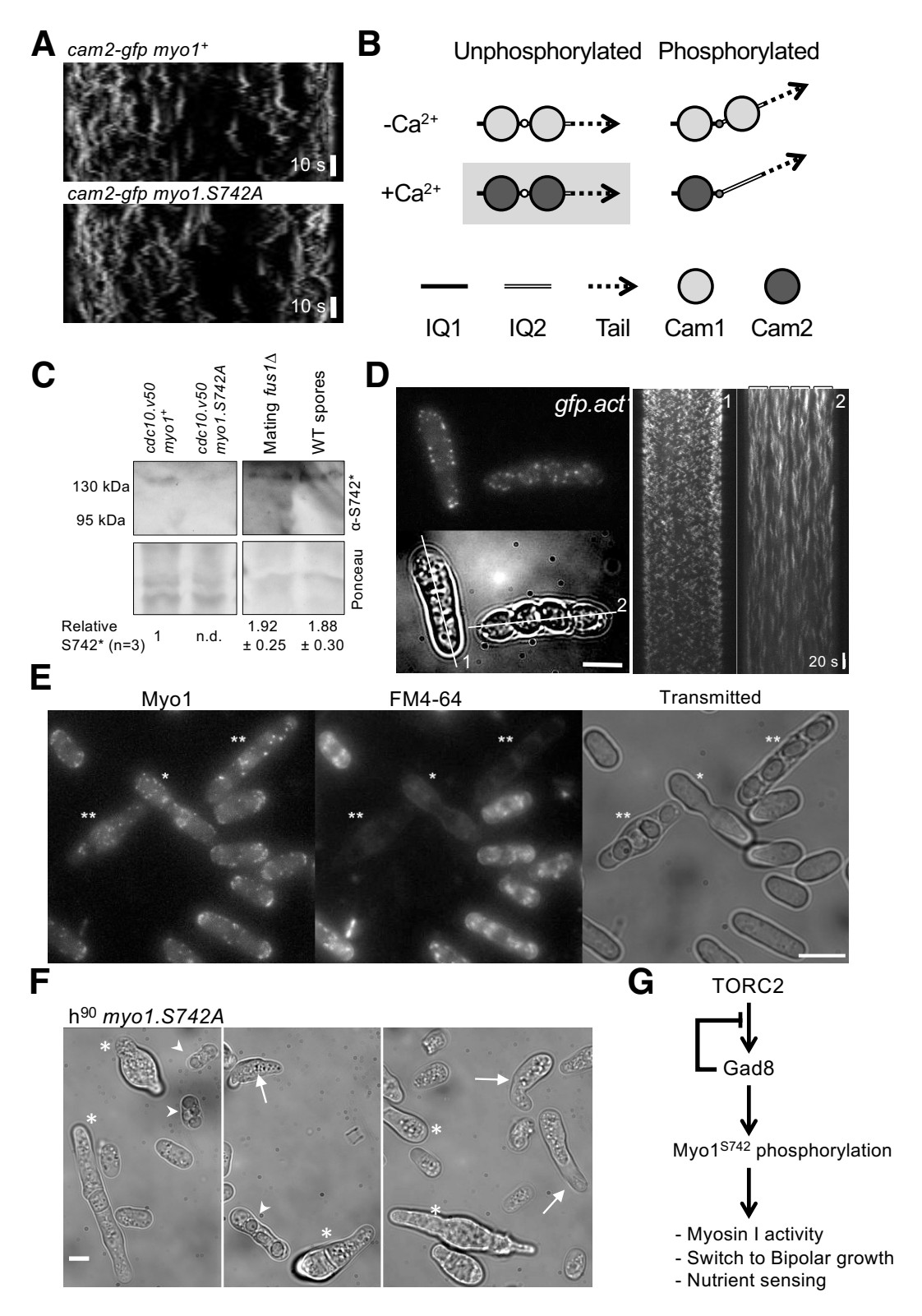

**Figure 8.** Myo1 S742 phosphorylation regulated Cam1 and Cam2 dynamics during meiosis. (**A**) Kymographs of Cam2.GFP foci dynamics in *myo1+* (upper panel) and *myo1.S742A* (lower panel) cells. (**B**) Scheme of the consequences of phosphorylation of Myo1[Ser742] (small empty circle) and Ca[2+] levels upon the binding of Cam1 (light gray filled circle) and Cam2 (dark gray filled circle) to the IQ1 (solid thick black line) and IQ2 (double line) motifs of Myo1, and the impact on the relative orientation of the myosin lever arm (dashed arrow). The highlighted combination of unphosphorylated

*Figure 8 continued on next page*

*Figure 8 continued*

Myo1$^{S742}$ and Ca$^{2+}$ does not normally occur in cells. (**C**) Western blots of extracts from G$_1$-arrested *cdc10.v50 myo1$^+$*, *cdc10.v50 myo1-S742A*, and conjugation arrested (starved, premeiotic cells) *fus1Δ* cells, and from meiotic spores, probed with phospho-specific anti-Myo1$^{S742}$ antibodies (upper panel). Ponceau staining (lower panel) confirms that Myo1S742 remains phosphorylated from the end of G$_1$, through conjugation until the end of meiosis (n = 5). Relative Myo1$^{S742}$ phosphorylation levels were calculated from three independent equivalent experiments (mean ± sd). (**D**) Left panel: maximum projection of a 13-z slice GFP fluorescence image (top) and a transmitted light image (bottom) from a time-lapse of vegetative (cell 1) and meiotic (cell 2) *gfp-act1* cells. Image from a GFP-act signal. The kymographs in the right panels were generated along the two dotted axes. (**E**) Maximum projection of mNeongreen-Myo1 fluorescence (left), FM4-64 fluoresence (middle) and transmitted light images of a mixed population of vegetative, fusing (*) and sporulating (**) *mNG-myo1$^+$* cells, illustrating that endocytosis is reduced in meiotic cells. Scale bar, 10 μm. (**F**) Micrographs illustrating *myo1.S742A* cell morphology on solid starvation medium. Asterisks highlight cells with unregulated growth and polarity defects; arrows highlight cells with elongated or abnormally bent shmooing (conjugation) tips; arrow heads highlight meiotic cells with defective spore formation. Scale bar, 5 μm. (**G**) A schematic of the TORC2–Gad8–Myo1$^{S742}$ signaling pathway.

DOI: https://doi.org/10.7554/eLife.51150.017

The number of Myo1 molecules at the plasma membrane focus remains constant as the membrane is internalized until 2 s after Cam1 disassociates from Myo1. The trigger for Cam1 release is unknown, but the speed at which the event takes place indicates that it is likely to be initiated by a rapid localized spike in calcium. This could perhaps be driven by a critical level of membrane deformation coupled to calcium influx, similar to processes proposed for mechano-transduction and the role of mammalian myosin-1 within the stereocilia of the inner ear (*Adamek et al., 2008*; *Batters et al., 2004*).

Once Cam1 detaches from the Myo1 molecule, the neck loses rigidity, reducing tension between the myosin motor and the actin filament, to promote detachment from F-actin (*Lewis et al., 2012*; *Mentes et al., 2018*). Single molecules of Myo1 do not reside for long at the plasma membrane (off rate is ~8 sec$^{-1}$, *Video 1*), so without an interaction with actin, Myo1 would leave the endocytic patch a second or so after losing its Cam1 light chain. Therefore, after Cam1 release, there is a 2 s delay in the disappearance of Myo1 signal as it disassociates from the endocytic machinery (*Figure 3C, D*).

The conformation and rigidity of the Myo1 lever arm therefore play a key role in modulating the tension-sensing properties of the motor domain. This is consistent with our data showing that wild-type phosphorylation-competent Myo1 resides at the membrane ~1.8 s longer than does Myo1 mutant protein that cannot be phosphorylated at serine 742 (Myo1$^{S742A}$) (*Figure 3—figure supplement 1*). Phosphorylation-dependent changes in the conformation of the myosin neck provide a simple mechanism to modulate the rate of endocytosis according to the size and needs of the cell. Similarly, in the presence of Ca$^{2+}$ and Myo1$^{S742}$ phosphorylation, a single Cam2 resides at the IQ1 motif of the neck (*Figure 8B*). While bringing about a change in the conformation of the first half of the myosin lever arm (adjacent to the motor domain), the vacant IQ2 motif allows flexibility within the carboxyl half of the neck region. This would provide a relatively tension-insensitive motor that stalls against the actin polymer, which would therefore persist significantly longer at the endocytic foci, as observed to occur here in meiotic cells (*Figure 8D*, *Supplementary file 1* Table 1). These changes in lever-arm properties change the overall rate of endocytosis, as observed in differences in the time taken for endosomes to internalize within the cytoplasm (*Supplementary file 1* Tables 1 and 2).

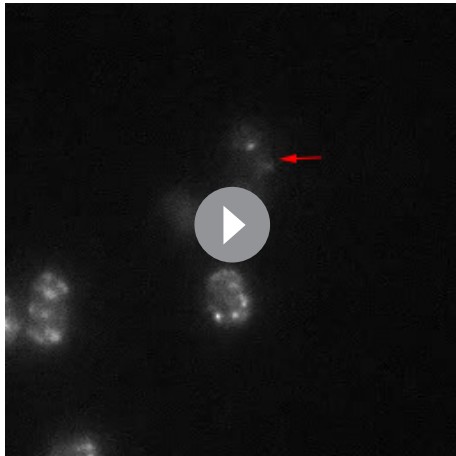

**Video 5.** Myo1 dynamics in interphase and meiotic cells. Time-lapse of maximum projections from 13-z slice widefield images of *mNeongreen.myo1* cells showing typical examples of Myo1 dynamics in vegetative and meiotic (highlighted with arrow) cells. Frame rate: 650 msec/frame.
DOI: https://doi.org/10.7554/eLife.51150.018

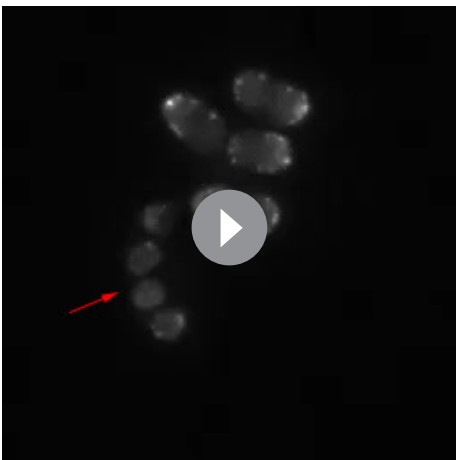

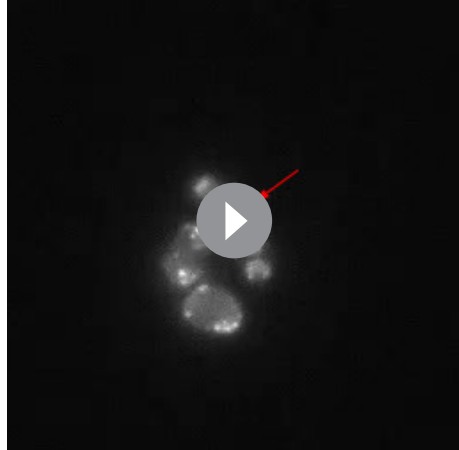

**Video 6.** Cam1 dynamics in interphase and meiotic cells. Time-lapse of maximum projections from 13-z slice widefield images of *cam1.gfp* cells showing typical examples of Cam1 dynamics in vegetative and meiotic (highlighted with arrow) cells. Frame rate: 650 msec/frame.
DOI: https://doi.org/10.7554/eLife.51150.019

**Video 7.** Cam2 dynamics in interphase and meiotic cells. Time-lapse of maximum projections from 13-z slice widefield images of *cam2.gfp* cells showing typical examples of Cam2 dynamics in vegetative and meiotic (highlighted with arrow) cells. Frame rate: 650 msec/frame.
DOI: https://doi.org/10.7554/eLife.51150.020

Thus, phosphorylation-dependent changes in the calcium-regulated conformation and rigidity of the myosin lever arm could provide a universal mechanism for regulating the diverse cytoplasmic activities and functions of myosin motors within all cells.

## Materials and methods

### Yeast cell culture

Cell culture and maintenance were carried out according to *Moreno et al. (1991)* using Edinburgh minimal medium with glutamic acid nitrogen source (EMMG) unless specified otherwise. Cells were cultured at 25°C unless stated otherwise and cells were maintained as early to mid-log phase cultures for 48 hr before being used for analyses. Genetic crosses were undertaken on MSA plates (*Egel et al., 1994*). All strains used in this study were prototrophs. They are listed in *Supplementary file 1*.

### Molecular biology

$cam1^+$ (SPAC3A12.14), *cam1.T6C* and $cam2^+$ (SPAC29A4.05) genes were amplified as *Nde1* - *BamH1* fragments from genomic *S. pombe* DNA using o226/o227 and o393/o394 primers and cloned into pGEM-T-Easy (Promega, Madison, WI, USA). After sequencing, the subsequent genes were cloned into pJC20 (*Clos and Brandau, 1994*) to generate bacterial calmodulin-expression constructs. DNA encoding for the FRET optimized fluorophores CyPet and YPet (*Nguyen and Daugherty, 2005*) were each amplified using primers o405/o406 and o403/o404, respectively. o406 also incorporated DNA at the 3' end of the CyPet ORF that encodes the

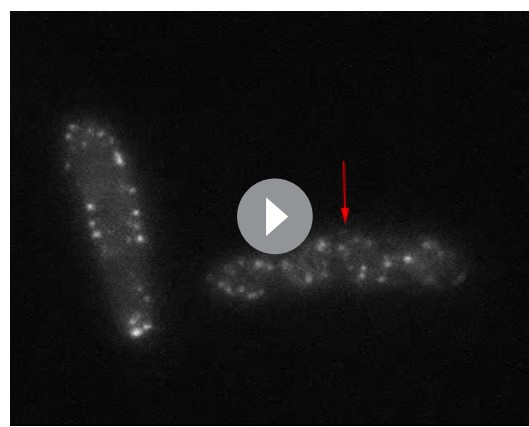

**Video 8.** Act1 dynamics in interphase and meiotic cells. Time-lapse of maximum projections from 13-z slice widefield images of *gfp.act1* cells showing typical examples of Act1 dynamics in vegetative and meiotic (highlighted with arrow) cells. Frame rate: 650 msec/frame.
DOI: https://doi.org/10.7554/eLife.51150.021

first IQ motif of the Myo1 neck region, whereas o404 included DNA encoding a Gly3His6 tag at the 3′ of the YPet ORF. The two DNA fragments were cloned into pGEM-T-Easy in a three-way ligation reaction to generate pGEM-CyPet–Myo1IQ1–YPet. The CyPet–Myo1$^{IQ1-}$YPet DNA was subsequently sequenced and cloned as a Nde1 - BamH1 fragment into pJC20 (*Clos and Brandau, 1994*) to generate pJC20CyPet–Myo1$^{IQ1}$–YPet. Complementary oligonucleotides o425 and o426 were annealed together and ligated into BglII – Xho1 cut pJC20CyPet–Myo1$^{IQ1}$–YPet to generate pJC20CyPet–Myo1$^{IQ12}$–YPet. Similarly, the complementary oligonucleotides o429 and o430 were annealed together and subsequently ligated into Sal1-BglII cut pJC20CyPet–Myo1$^{IQ1}$–YPet and the subsequent Xho1 fragment was excised to generate pJC20CyPet–Myo1$^{IQ2}$–YPet. Site-directed mutagenesis was carried out using the pJC20CyPet–Myo1$^{IQ12}$–YPet template and o427 and o428 primers to generate pJC20CyPet–Myo1$^{IQ12}$S742D–YPet. Complementary oligonucleotides o449 and o450 were annealed together and ligated into Nru1 – Xho1 digested pJC20CyPet–Myo1$^{IQ12}$S742D–YPet to generate pJC20CyPet–Myo1$^{IQ12}$S742A–YPet. All plasmids were sequenced upon construction. Strains with fluorophore-tagged alleles of *cam1$^+$* and *cam2$^+$* were generated as described previously using appropriate templates and primers (*Bähler et al., 1998*). Strains in which the *myo1.S742A*, *myo1.S742D*, *mNeongreen-myo1*, *mNeongreen-myo1.S742A*, or *mNeongreen-myo1.S742D* alleles replaced the endogenous *myo1$^+$* gene (SPBC146.13c) were generated using a marker switching method (*MacIver et al., 2003*). Oligonucleotides are described in *Supplementary file 2*.

## Protein expression and purification

All recombinant proteins were expressed and purified from BL21 DE3 *Escherichia coli* cells, except Cam1 proteins for which BL21 DE3 pNatA cells (*Eastwood et al., 2017*) were used to allow amino-terminal acetylation (*Figure 2—figure supplement 1*). For calmodulin purification, cell lysates were resuspended in Buffer A (50 mM Tris, 2 mM EDTA, 1 mM DTT, 0.1 mM PMSF (pH 7.5)) and pre-cleared by high-speed centrifugation (48,500 RCF; 30 min; 4°C), before ammonium sulphate was added to the supernatant at 35% saturation, and the mixture incubated for 30 min at 4°C. Precipitated proteins were removed by centrifugation (48,500 RCF; 30 min; 4°C). For Cam1 purifications, the precipitation-cleared supernatant was added to a pre-equilibrated 10 ml phenyl sepharose (CL-4B) column (Buffer B: 50 mM Tris, 1 mM DTT, 1 mM NaN$_3$, 5 mM CaCl$_2$ (pH 8.0)), then washed in four volumes of Buffer B before being eluted as fractions in Buffer C (50 mM Tris, 1 mM DTT, 1 mM NaN$_3$, 5 mM EGTA (pH 8.0)). For Cam2 purification, the precipitation-cleared supernatant underwent a second round of ammonium sulphate precipitation and clearing, and the subsequent supernatant was subjected to isoelectric precipitation (pH 4.3) and centrifugation (48,500 RCF: 30 min; 4°C). The resultant pellet was resuspended in Buffer A and heated to 80°C for 5 min, before denatured proteins were removed by centrifugation (16,000 RCF; 5 min). *His-tagged* proteins were purified in native conditions using prepacked, pre-equilibrated 5 ml Ni$^{2+}$ columns.

## Immunological techniques

Standard immunological methods were used as described (*Harlow and Lane, 1988*). Serine 742 phosphorylation-state specific anti-Myo1 antibodies were raised against phosphate-conjugated peptide encompassing Myo1 serine 742 in SPF rabbits (Eurogentec, Seraing, Belgium). These antibodies were subsequently affinity-purified.

## Analysis of yeast extracts

Protein extracts were prepared and analyzed as described elsewhere (*Baker et al., 2016*). For western blot analysis, anti-Myo1 sera was diluted 1:1000, whereas Myo1 serine 742 phosphorylation state specific antibodies were used at a dilution of 1:50. Gel densitometry was undertaken using ImageJ software.

## Fast reaction kinetics

All transient kinetics were carried out using a HiTech Scientific DF-61 DX2 Stopped Flow apparatus (TgK Scientific, Bradford-upon-Avon, UK) at 20°C. All data were acquired as the average of 3–5 consecutive shots and analyzed using the KineticStudio software supplied with the equipment. Quin-2 fluorescence was excited at 333 nm and a Schott GG445 cut off filter was used to monitor fluorescence above 445 nm. IAANS (2-(4′-(iodoacetamido)anilino)-naphthalene-6-sulfonic acid) was excited

at 335 nm and fluorescence was monitored through a GG455 filter. For the FRET measurements, CyPet was excited at 435 nm and YPet emission was monitored through a combination of a Wrattan Gelatin No12 (Kodak) filter with a Schott GG495 nm filter to monitor fluorescence at 525–530 nm.

## Fluorescence spectra

Emission spectra were obtained using a Varian Cary Eclipse Fluorescence Spectrophotometer (Agilent Technologies, Santa Clara, CA) using a 100 µl quartz cuvette. For FRET measurements, samples were excited at 435 nm (CyPet excitation) and emission was monitored from 450 to 600 nm with both slits set to 1 nm. Affinity experiments were carried out using 1 µM IQ-FRET protein with varying concentrations of Cam1 or Cam2 in a final volume of 100 µl in analysis buffer of 140 mM KCl, 20 mM MOPS (pH 7.0), with or without 2 mM $MgCl_2$ and with 2 mM of EGTA, $CaCl_2$ or $Ca^{2+}$-EGTA as required. Distances between FRET fluorophores were calculated as described previously (*Wu and Brand, 1994*) using an CyPet–YPet $R_0$ value of 53.01.

## Live cell imaging

Live cell widefield fluorescence imaging was undertaken as described previously (*Baker et al., 2016*). For total internal reflection fluorescence microscopy (TIRFM), *S. pombe* cells were immobilized on №1, Ø 25 mm lectin-coated coverslips and placed into imaging chambers filled with EMMG medium. A previously described custom TIRF microscope (*Mashanov et al., 2003*) was used to image individual cells at a rate of 20 fps in either single of dual color mode. Lasers were 488 nm/100 mW and 561 nm/150 mW (Omicron, Germany); emission filters 525/50 nm and 585/29 nm; dichroic mirror 552 nm (Semrock, NY); all other lenses and mirrors were from Thorlabs (NJ), except two Ø3 mm mirrors (Comar Optics, UK) which directed light in and out of the 100 × 1.45 NA objective lens (Olympus, Japan). Sequences of images were captured using one or two iXon897BV cameras (Andor Technology, UK) with custom-made acquisition software. 100% laser power (488 nm) was used to image individual mNeongreen–Myo1 and Cam1–GFP molecules. The laser intensity was reduced to 20% during endocytosis imaging experiments to minimize photobleaching. All imaging was undertaken at 23 ℃.

## Immunofluorescence

Immunofluorescence microscopy was performed as described previously (*Hagan and Hyams, 1988*), except gluteraldehyde was omitted. Images were captured using the above widefield imaging system. Anti-Myo1 sera (*Attanapola et al., 2009*) were used at a dilution of 1:100, whereas affinity-purified Myo1 serine 742 phosphorylation state-specific antibodies were used at a dilution of 1:10.

## Image analysis

Widefield data were analyzed using Autoquant software (MediaCybernetics, Rockville, MD). All 3D image stacks were subjected to blind 3D deconvolution before analysis. Average size, number and cellular distribution of foci were calculated from all foci present within ≥30 cells for each sample examined. Timing of foci events were calculated from kymographs generated in Metamorph software (Molecular Devices, Sunnyvale, CA). The proportion of cells displaying a bent cell phenotype was determined from more than 350 calcofluor (1 mg.ml$^{-1}$) stained cells for each strain. Bent cells were defined by a deviation in the direction of growth of >5˚ from the longitudinal axis.

TIRF data analyses, including single-molecule detection and tracking, were undertaken using GMimPro software (*Mashanov and Molloy, 2007*). Endocytic events were identified by creating an image representing the standard deviation of each pixel over the whole video sequence (known as a 'z-projection'). Bright spots in this image correspond to regions of the yeast cell that showed large intensity fluctuations associated with endocytic activity. Intensity trajectories in these regions of interest (ROIs) (0.5 µm diameter, 5 × 5 pixels) were saved for future analysis. To correct for local variation in background signal, the average intensity in a region of 1.5 µm diameter around the endocytosis site (but not including the central region of interest) was subtracted. Data from ROIs that were contaminated by other endocytosis events, occurring in close proximity and close in time, were manually excluded from the analysis. It was critical to identify the start and end of each endocytosis event accurately so that individual traces could be averaged. To facilitate this, the rising and falling phases of the intensity trace were fitted with a straight line (60 data points, 3 s duration), see *Figure 3C* for

example. The intercept of this line with the baseline intensity gave the $t_{start}$ and $t_{end}$ values and event duration ($T_{dur} = T_{end}$ $Tst_{art}$) (see *Figure 7A*). The amplitude (intensity) of the event $A_{av}$ was measured at the middle of the event by averaging 60 data points from 5$^{th}$ to 8$^{th}$ second from the $T_{start}$ (gray bar on *Figure 3B*). Intensity traces for each given condition were synchronized to the starting point ($T_{start}$) and averaged. (Except Cam2-GFP traces, which were synchronized using $t_{start}$ measured from simultaneously acquired Cam1-mCherry signal.) Similarly, traces were synchronized to their end point ($T_{end}$) and averaged. The mean duration of the events ($T_{dur}$) for each condition was then used to reconstruct the mean intensity changes with calculated errors for event amplitude and timing (*Supplementary file 1* Table 2). We used the results of single-molecule imaging experiments to calculate the number of single fluorescent molecules contributing to the spot intensity at a given time. As the falling and rising phases of most events fitted well to a simple linear equation, the slope of the fitted lines was used to estimate the rate of accumulation and dissociation of the fluorescent molecules (*Figure 3C*). As Cam2-GFP remained bound to the endocytic vesicle, when vesicle scission occurred, intensity fell rapidly to zero as the vesicle diffused from the TIRF evanescent field; the time of scission was defined as $T_{scis}$ (*Figure 7A*). Single-particle tracking was performed using GMimPro (*Mashanov and Molloy, 2007*) (ASPT module) so that the paths (or trajectories) of individual Myo1 molecules bound to cell membrane could be traced. Trajectories were analyzed to yield mean intensities for individual mNeonGreen- and eGFP-labeled proteins, which could be used to estimate the number of fluorescently tagged molecules associated with each endocytotic event. Intensity-versus-time plots were generated from averages of >30 foci for each protein in each genetic background examined.

## Acknowledgements

We thank M Balasubramanian, I Hagan, P Nurse, C Shimoda and T Pollard for strains; and Ben Goult, Iain Hagan and Janni Petersen for stimulating discussions and comments on the manuscript. This work was supported by the University of Kent and by funding from the Biotechnology and Biological Sciences Research Council (BB/J012793/1 and BB/M015130/1), by a Royal Society Industry Fellowship to DPM; by a CASE industrial bursary from Cairn Research Ltd to KB and by the Francis Crick Institute, which receives core funding from Cancer Research UK (FC001119), the UK Medical Research Council (FC001119) and the Wellcome Trust (FC001119 to GIM and JEM).

## Additional information

### Funding

| Funder | Grant reference number | Author |
|---|---|---|
| Biotechnology and Biological Sciences Research Council | BB/J012793/1 | Michael A Geeves<br>Daniel P Mulvihill |
| Biotechnology and Biological Sciences Research Council | BB/M015130/1 | Daniel P Mulvihill |
| Royal Society | Industry Fellowship | Daniel P Mulvihill |
| Cancer Research UK | FC001119 | Gregory I Mashanov<br>Justin E Molloy |
| Medical Research Council | FC001119 | Gregory I Mashanov<br>Justin E Molloy |
| Wellcome | FC001119 | Gregory I Mashanov<br>Justin E Molloy |
| Cairn Research Ltd, Faversham, UK | iCASE industrial bursary | Karen Baker |

The funders had no role in study design, data collection and interpretation, or the decision to submit the work for publication.

## Author contributions

Karen Baker, Data curation, Formal analysis, Investigation, Writing—review and editing; Irene A Gyamfi, Formal analysis, Investigation; Gregory I Mashanov, Software, Investigation, Methodology, Writing—review and editing; Justin E Molloy, Formal analysis, Supervision, Writing—review and editing; Michael A Geeves, Conceptualization, Supervision, Methodology, Project administration, Writing—review and editing; Daniel P Mulvihill, Conceptualization, Data curation, Formal analysis, Supervision, Funding acquisition, Methodology, Writing—original draft, Project administration, Writing—review and editing

## Author ORCIDs

Justin E Molloy (iD) http://orcid.org/0000-0002-8307-2450
Michael A Geeves (iD) http://orcid.org/0000-0002-9364-8898
Daniel P Mulvihill (iD) https://orcid.org/0000-0003-2502-5274

## Decision letter and Author response

Decision letter https://doi.org/10.7554/eLife.51150.028
Author response https://doi.org/10.7554/eLife.51150.029

## Additional files

### Supplementary files

• Supplementary file 1. Strains used during this study.
DOI: https://doi.org/10.7554/eLife.51150.022

• Supplementary file 2. Oligonucleotides used during this study.
DOI: https://doi.org/10.7554/eLife.51150.023

• Transparent reporting form DOI: https://doi.org/10.7554/eLife.51150.024

### Data availability

Raw data files for Figures and Tables, and data analysis spreadsheets, are uploaded onto the University of Kent Data Repository server and are available at the following location: https://data.kent.ac.uk/60/.

The following dataset was generated:

| Author(s) | Year | Dataset title | Dataset URL | Database and Identifier |
|---|---|---|---|---|
| Mulvihill DP | 2010 | TORC2 dependent phosphorylation modulates calcium regulation of fission yeast myosin | https://data.kent.ac.uk/60/ | Kent Data Repository, 60 |

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
