## [Decision Letter]

[Editors’ note: a previous version of this study was rejected after peer review, but the authors submitted for reconsideration. The first decision letter after peer review is shown below.]

Thank you for submitting your work entitled "TORC2 dependent phosphorylation modulates calcium regulation of fission yeast myosin" for consideration by *eLife*. Your article has been reviewed by three peer reviewers, one of whom is a member of our Board of Reviewing Editors, and the evaluation has been overseen by a Senior Editor. The reviewers have opted to remain anonymous.

Our decision has been reached after consultation between the reviewers. Based on these discussions and the individual reviews below, we regret to inform you that your work will not be considered further for publication in *eLife*, at least in its present form.

This manuscript reports identification of a novel mechanism by which phosphorylation of fission yeast myosin-1 heavy chain, in combination with calponins, controls endocytosis and actin dynamics during bipolar growth and meiosis. The three reviewers stated that the data presented in the manuscript are interesting and potentially very important. However, the manuscript has several shortcomings. First, the manuscript organization is confusing, and it lacks many important details making it very hard to follow. Second, some experiments (e.g. the blots presented in Figure 1) are of insufficient quality, and should be repeated. Third, the link between myosin-1 phosphorylation and TORC2 should be examined in much more detail, and it would be important to provide better insights into physiological roles of TORC2-dependent phosphorylation of myosin-1. Thus, an extensive amount of additional work would be required to address these issues.

All three reviewers, however, felt that the results presented in the manuscript are novel and interesting. Therefore, if you can address the main points raised by the reviewers by extensively re-writing the manuscript, by carrying out much more thorough analysis of the TORC2-myosin-1 interplay (i.e. provide stronger evidence that phosphorylation at this site in myosin-1 is TORC2-dependent, and examine whether it is a direct substrate of TORC2 or an indirect substrate of a protein kinase that is activated via TORC2-mediated phosphorylation), as well as by providing better quality data on certain key experiments, we would be glad to consider a new submission on this topic for publication in *eLife*. We hope that the comments by the three reviewers will help you in improving the manuscript, and in deciding whether you will resubmit this work to *eLife* or elsewhere.

*Reviewer #1:*

This manuscript focuses on the mechanisms by which phosphorylation and calmodulins control the activity of fission yeast myosin-1 in vitro and in cells. The authors demonstrate that the *S. pombe* myosin-1 can be phosphorylated within the IQ motif in a TORC2-dependent manner. This myosin phosphorylation peaks at the early interphase during vegetative growth as well as from G1 until completion in spore formation during meiosis. Replacing the wild-type myosin-1 by a phospho-deficient *myo1.S742A* mutant results in various abnormalities, including defects in actin dynamics at the old cell end during bipolar growth as well as in various deficiencies during meiosis. By extensive biochemical and cell biological analysis, the authors present evidence that only one calmodulin isoform, Cam1, associates with myosin-1 during vegetative growth. They show that phosphorylation at S742 does not affect Cam1-binding to myosin-1, but instead modulates lever arm length upon Cam1 binding. On the other hand, both Cam1 and Cam2 associate with myosin-1 during meiosis, and here the association of Cam1 and Cam2 with myosin-1 appears to be affected by S742 phosphorylation.

The manuscript contains an impressive amount of data, and it provides interesting new insights into the interplay between myosin-1, calmodulins, and TORC2 signaling pathway. However, the manuscript was quite difficult to follow for two reasons. First, the figure legends and the manuscript text lack many important details, thus making it often hard to understand the rationale and details of the performed experiments. Second, the manuscript organization was somewhat confusing and it appeared as loosely related observations on myosin-1 phosphorylation, calmodulins, endocytosis and meiosis. Thus, the manuscript would benefit from significant re-writing and re-organization, as well as on better 'Discussion' (and a model figure) that would more clearly summarize the main findings.

1) The physiological links between TORC2, myosin-2 phosphorylation, endocytosis, and meiosis remain somewhat elusive from the manuscript. Moreover, it remained unclear whether Ste20 (Rictor) phosphorylates myosin-1 at S742 or if myosin-1 is phosphorylated by some other downstream kinase. Thus, better discussion on the links between TORC2-dependent myosin-1 phosphorylation and the observed phenotypes would be required to address these issues.

2) The blots presented in Figure 1 are of insufficient quality. For example, from the blot presented in Figure 1A it is difficult the see the different myosin-1 bands over the background. Moreover, I do not understand why myosin-1 (as detected by a-*myo1* antibody) runs as a sharp band in *ste20*^+^ sample and as a smear in the *ste20Δ* sample. Finally, the data presented in Figure 1F should be carefully quantified to convincingly show the effects of *myo1.S742* on the growth inhibition in minimal nitrogen media.

3) The links between myosin-1 phosphorylation and calmodulins with abnormal shmoo tips and spore defect phenotype in meiosis are not particularly clear. Thus, some additional experiments could clarify this issue. The authors show in Figure 7D that actin foci are much less dynamic during meiosis compared to vegetative cells. This is interesting. Could the authors examine the possible effects of Cam2 deletion and *myo1.S742S* on the dynamics of actin patches during meiosis? Does Cam2 deletion affect myosin-1 or Cam1 dynamics during meiosis?

4) The model figure (Figure 8) is not particularly informative, and e.g. the panels A-E merely summarize the data from earlier publications (and not the new findings presented in this manuscript). Thus, the authors should prepare a completely new version of the model figure that summarizes their own findings on the effects of TORC2-dependent myosin-1 phosphorylation (and Ca^2+^) on Cam1 and Cam2 binding, on lever arm length, and on the function of myosin-1 during vegetative growth and meiosis.

*Reviewer #2:*

The manuscript by Baker et al. examines the potential novel regulation of yeast myosin-1 by phosphorylation of the heavy chain in one of the calmodulin binding sites. The study uses an interesting combination of biochemical and live cell imaging techniques to approach the problem. The study is novel and should be of interest to a broad audience from biochemists to cell biologist. I do have some concerns which should be addressed.

1) The manuscript is not well written and is hard to follow. This is, in part, due to lack of explanation in several places, figures being labeled with abbreviations that are not explained and lack of detail in the figure legends.

2) I worry that the use of an anti-phosphomyosin antibody is not going to address stoichiometry of phosphorylation. The authors never really show that phosphomyosin-1 is associating with endocytic patches. What if only 10% of the myosin ever gets phosphorylated and that this material remains in the cytoplasm? Can the anti-phosphomyosin antibody be used for immunofluorescent studies of the cells? In the absence of this, could you incubate the cells with phosphatase inhibitors and see whether the normalized staining of the *myo1* band increases beyond what you're currently seeing in control cells?

3) I'm always a bit suspicious of phosphomimetic mutations. For example, in vitro myosin-2 studies phosphomimetic replacement of S19 does not mimic the effects of phosphorylation very well as shown in a study from the Sweeney and Trybus labs in addition to others (e.g. Sweeney et al., Proc Natl Acad Sci, 91:1490−1494, 1994). Are any AGC family kinases available commercially and, if so, have the authors considered buying some and trying to phosphorylate the IQ peptides to determine its effect on calmodulin binding?

*Reviewer #3:*

In this work, the authors utilize a mutant allele (Myo1^S742A^) of the Type 1 myosin of *S. pombe* Myo1 to interrogate how presumed TORC2-dependent phosphorylation of Myo1 modulates its interaction with calmodulins (Cam1 and Cam2) and affects endocytosis at specific locations within the cell. Characterization of interaction of the Cam's with Myo1 by an in vitro binding assay (via intramolecular FRET) seems well done, convincing, and provides a strong foundation for interpreting the in vivo microscopy observations of both wild-type and the *myo1.S742A* mutant cells. However, of grave concern is the fact that the major claim of this paper – that Myo1 function is regulated via TORC2-dependent phosphorylation (as stated even in the Title of the paper) – is not substantiated. Given that the most novel aspect of this work is, purportedly, that Myo1 is regulated via its TORC2-mediated modification, publication of the work cannot be recommended unless and until more convincing data are presented to support this specific conclusion. In addition, and regrettably, the text of the manuscript overall is very convoluted and often lacks key explanations of important experimental details, which makes interpretation of the results difficult for the reader, and would need extensive revision before further consideration for publication in *eLife*.

1) The immunoblots presented in Figures 1A, 1C, and 7C are of extremely poor quality. Because the results of these particular experiments are the primary data used to argue that TORC2 phosphorylates Myo1 at S742, these studies need to be repeated to provide more convincing evidence. In this regard, the most highly phosphorylated species of Myo1 still seem to be present in the *ste20Δ* and *gad8Δ* mutant strains. This ambiguity can be addressed by treatment of the lysates with phosphatase to determine which bands are actually phosphorylated species. For Figure 1A, no mention is made in either the corresponding figure panel or manuscript text about the rationale behind the use of the *gad8Δ* strain.

2) It is clear that the *myo1.S742A* allele has a much more deleterious effect in cells undergoing meiosis (Figure 6) than in cells undergoing vegetative growth (Figure 4). If TORC2 does indeed phosphorylate Myo1, it would be important to know, first, whether TORC2 function is elevated in meiotic cells compared to mitotic cells and, second, what the effect of different growth conditions (vegetative, sexual, and nutrient limiting) is on the association of Myo1 with Cam1 vs. Cam2 as assessed via co-immunoprecipitations. These experiments would lend at least some support for a link between TORC2 and calcium signaling pathways in the regulation of Myo1. One prediction from the data presented in the present manuscript is that binding of Cam2 to Myo1 could regulate endocytic events to inhibit growth.

3) It is unclear why the authors chose to use only the non-phosphorylated *myo1.S742A* allele in all of the in vivo experiments and only the phosphomimetic *myo1.S742D* allele in all of the in vitro experiments. Both alleles should be used in under both assay conditions to provide essential internal consistency and appropriate controls in all of the experiments.

4) Figure 3 presents evidence suggesting that the *myo1.S742A* mutant strain undergoes atypical polar cell growth, but no model or explanation for this phenotype is given. Statistical analysis of the sub-cellular distribution of the mNeonGreen.Myo1 signals in Figure 4D and the distribution of actin may help to explain this observation. With regard to the misregulation of polar growth caused by the *myo1.S742A* allele, this phenotype does not seem very deleterious and certainly does not significantly affect the growth rate of the cells (Figure 4C). Can this allele be combined with any other genetic backgrounds that are known to affect polar growth to exacerbate this phenotype?

5) Figure 2 presumably contains a wealth of useful data, but is extremely hard to interpret due to a lack of detailed experimental information in the corresponding figure legend, as well as a general lack of labels and titles on the graphs (specifically panels B, C, and I).

[Editors’ note: what now follows is the decision letter after the authors submitted for further consideration]

Thank you for submitting your article "TORC2-Gad8 dependent myosin phosphorylation modulates regulation by calcium" for consideration by *eLife*. Your article has been reviewed by three peer reviewers, including Pekka Lappalainen as the Reviewing Editor and Reviewer #1, and the evaluation has been overseen by Anna Akhmanova as the Senior Editor.

The reviewers have discussed the reviews with one another and the Reviewing Editor has drafted this decision to help you prepare a revised submission.

Summary:

This is a resubmission of a manuscript focusing on the links between calcium, TORC2, and type 1 myosin (Myo1) in actin dynamics. The authors have carried out several new experiments, as well as revised the manuscript text according to the suggestions by three original reviewers. The study provides interesting new information on the roles of fission yeast calmodulins (Cam1 and Cam2), and TORC2/Gad8 dependent phosphorylation on the activity and cellular functions of Myo1.

The current version of the manuscript was evaluated by two of the original reviewers who found the manuscript significantly improved. Because one of the original reviewers was unable to evaluate the new version of the manuscript, it was also reviewed by another expert on TOR signaling. This reviewer stated that some additional experiments are required to strengthen the conclusions concerning the TORC2/Gad8 dependence of Myo1 phosphorylation, and its cell cycle regulation. Finally, the manuscript is not particularly reader-friendly, and the authors should still revise the manuscript text to make it better accessible for a non-specialist reader.

Essential revisions:

1) The authors show that Myo1^S742^ phosphorylation is dependent on Gad8 and Ste20 (rictor subunit in TORC2). However, whether this is a direct or indirect phosphorylation remains unknown. The data would be more convincing if the authors would replace Figure 1B with a better quality blot, more rigorously quantify the levels of phosphorylated Myo1, and also include a *tor1Δ* mutant strain to their analysis. In the absence of biochemical experiments to test Myo1 phosphorylation by Gad8, they should clearly state that the data do not necessarily support direct phosphorylation of Myo1 by Gad8.

2) For the studies on cell cycle regulation of Gad8 and Myo1^S742^-P, the authors should also test if the level of TORC2-Gad8 activity is regulated in a cell cycle dependent manner. If this is not the case, the authors should tone down their conclusions that Gad8 is responsible for the cell cycle phosphorylation of Myo1.

---

## [Author Response]

[Editors’ note: the author responses to the first round of peer review follow.]

Reviewer #1:[…] The manuscript contains an impressive amount of data, and it provides interesting new insights into the interplay between myosin-1, calmodulins, and TORC2 signaling pathway. However, the manuscript was quite difficult to follow for two reasons. First, the figure legends and the manuscript text lack many important details, thus making it often hard to understand the rationale and details of the performed experiments. Second, the manuscript organization was somewhat confusing and it appeared as loosely related observations on myosin-1 phosphorylation, calmodulins, endocytosis and meiosis. Thus, the manuscript would benefit from significant re-writing and re-organization, as well as on better 'Discussion' (and a model figure) that would more clearly summarize the main findings.1) The physiological links between TORC2, myosin-2 phosphorylation, endocytosis, and meiosis remain somewhat elusive from the manuscript. Moreover, it remained unclear whether Ste20 (Rictor) phosphorylates myosin-1 at S742 or if myosin-1 is phosphorylated by some other downstream kinase. Thus, better discussion on the links between TORC2-dependent myosin-1 phosphorylation and the observed phenotypes would be required to address these issues.

Since the initial submission, we have looked with more detail into the TORC2 dependent Myo1^S742^ phosphorylation, and have new evidence demonstrating Gad8, an AGC kinase that functions downstream of TORC2, as the protein kinase responsible for Myo1^S742^ phosphorylation. This is consistent with the serine location within an AGC consensus site, the reported cell cycle variation in Gad8 kinase activity in the mitotic and meiotic cell cycles and Gad8’s role in regulating the switch from monopolar to bipolar. We have added new data showing Myo1^S742^ phosphorylation is abolished in cells lacking Gad8 (Figure 1B), and that it is significantly reduced in *gad8.T6D* cells (Figure 1B), which have significantly reduced kinase activity compared to the wild type. We have added more background text in the manuscript and models to Figures 1 and 7 to provide the reader the appropriate context of how the study relates to the TORC2-Gad8 signalling pathway.

We looked into attempting in vitro kinase assays with the endogenous fission protein kinase to provide further evidence of direct phosphorylation. Unfortunately none of the epitope labelled *gad8* strains are functional, and each display growth and cell cycle defects.

2) The blots presented in Figure 1 are of insufficient quality. For example, from the blot presented in Figure 1A it is difficult the see the different myosin-1 bands over the background. Moreover, I do not understand why myosin-1 (as detected by a-myo1 antibody) runs as a sharp band in ste20^+^ sample and as a smear in the ste20Δ sample. Finally, the data presented in Figure 1F should be carefully quantified to convincingly show the effects of myo1.S742 on the growth inhibition in minimal nitrogen media.

We have repeated these westerns and replaced each of the blots shown in Figure 1. Statistics for cell lengths of cells shown in the new Figure 1E (old Figure 1F) have been added to the text.

3) The links between myosin-1 phosphorylation and calmodulins with abnormal shmoo tips and spore defect phenotype in meiosis are not particularly clear. Thus, some additional experiments could clarify this issue. The authors show in Figure 7D that actin foci are much less dynamic during meiosis compared to vegetative cells. This is interesting. Could the authors examine the possible effects of Cam2 deletion and myo1.S742S on the dynamics of actin patches during meiosis? Does Cam2 deletion affect myosin-1 or Cam1 dynamics during meiosis?

We have now undertaken timelapse imaging of GFP-Act1 in *myo1.S742A*, as well as *mNG.myo1* and *cam1.gfp* in and *cam2∆* cells in meiosis, and the data is included in Table 1.

4) The model figure (Figure 8) is not particularly informative, and e.g. the panels A-E merely summarize the data from earlier publications (and not the new findings presented in this manuscript). Thus, the authors should prepare a completely new version of the model figure that summarizes their own findings on the effects of TORC2-dependent myosin-1 phosphorylation (and Ca^2+^) on Cam1 and Cam2 binding, on lever arm length, and on the function of myosin-1 during vegetative growth and meiosis.

We have now removed Figure 8.

Reviewer #2:The manuscript by Baker et al. examines the potential novel regulation of yeast myosin-1 by phosphorylation of the heavy chain in one of the calmodulin binding sites. The study uses an interesting combination of biochemical and live cell imaging techniques to approach the problem. The study is novel and should be of interest to a broad audience from biochemists to cell biologist. I do have some concerns which should be addressed.1) The manuscript is not well written and is hard to follow. This is, in part, due to lack of explanation in several places, figures being labeled with abbreviations that are not explained and lack of detail in the figure legends.

The manuscript has been changed and edited significantly, to make it easier to read and follow.

2) I worry that the use of an anti-phosphomyosin antibody is not going to address stoichiometry of phosphorylation. The authors never really show that phosphomyosin-1 is associating with endocytic patches. What if only 10% of the myosin ever gets phosphorylated and that this material remains in the cytoplasm? Can the anti-phosphomyosin antibody be used for immunofluorescent studies of the cells? In the absence of this, could you incubate the cells with phosphatase inhibitors and see whether the normalized staining of the myo1 band increases beyond what you're currently seeing in control cells?

We have now established conditions for undertaking immunofluorescence with the anti-Myo1 S742-phosphorylated-antibody, and this illustrates the phosphorylated form of the Myo1 protein localises to at least a sub-set of the endocytic Myo1 foci (see Figure 3A).

3) I'm always a bit suspicious of phosphomimetic mutations. For example, in vitro myosin 2 studies phosphomimetic replacement of S19 does not mimic the effects of phosphorylation very well as shown in a study from the Sweeney and Trybus labs in addition to others (e.g. Sweeney et al., Proc Natl Acad Sci, 91:1490−1494, 1994). Are any AGC family kinases available commercially and, if so, have the authors considered buying some and trying to phosphorylate the IQ peptides to determine its effect on calmodulin binding?

While we appreciate the reviewer’s reservations on the use of phosphomimetic substitutions, we are confident in the phosphomimetic nature of the proteins, as the data is consistent with the in vivo findings. We could find no commercially produced full length equivalent AGC kinases available. We also looked into attempting in vitro kinase assays with the endogenous fission Gad8 kinase to provide further evidence of direct phosphorylation.

However none of the epitope labelled *gad8* (or Tor1) strains, from which the Gad8 would be immunoprecipitated, are functional, each displaying growth and cell cycle defects.

Reviewer #3:[…]1) The immunoblots presented in Figures 1A, 1C, and 7C are of extremely poor quality. Because the results of these particular experiments are the primary data used to argue that TORC2 phosphorylates Myo1 at S742, these studies need to be repeated to provide more convincing evidence. In this regard, the most highly phosphorylated species of Myo1 still seem to be present in the ste20Δ and gad8Δ mutant strains. This ambiguity can be addressed by treatment of the lysates with phosphatase to determine which bands are actually phosphorylated species. For Figure 1A, no mention is made in either the corresponding figure panel or manuscript text about the rationale behind the use of the gad8Δ strain.

We have repeated these westerns and replaced each of the blots shown in Figure 1. We have repeated the experiment presented in the original Figure 7C, and due to protein instability in starved cells, we have not been able to improve on the original blot presented, however, we feel it supports the statement that Myo1 serine 742 is phosphorylated during the sexual/meiotic cycle.

We have added text to describe the rationale behind the *gad8∆* strain and added additional data that address the ambiguity – see response to reviewer 1.

2) It is clear that the myo1.S742A allele has a much more deleterious effect in cells undergoing meiosis (Figure 6) than in cells undergoing vegetative growth (Figure 4). If TORC2 does indeed phosphorylate Myo1, it would be important to know, first, whether TORC2 function is elevated in meiotic cells compared to mitotic cells and, second, what the effect of different growth conditions (vegetative, sexual, and nutrient limiting) is on the association of Myo1 with Cam1 vs. Cam2 as assessed via co-immunoprecipitations. These experiments would lend at least some support for a link between TORC2 and calcium signaling pathways in the regulation of Myo1. One prediction from the data presented in the present manuscript is that binding of Cam2 to Myo1 could regulate endocytic events to inhibit growth.

We have now clarified that Gad8 is the kinase responsible for phosphorylating Myo1, and its published activity coincides with the events activity in the mitotic and meiotic cell cycles, and roles in promoting the switch to bipolar growth and responding to nutrient abundance coincides precisely with our observations on the serine 742 phosphorylation dependent Myo1 functions.

While we were are able to immunoprecipitate Myo1, Cam1, or Cam2 alone, we were unable to detect co-purification of the other proteins (not shown). Other publications that have demonstrated co-immunoprecipitation of fission yeast Myo1 with calmodulin have done so using strains in which the heavy and/or light chains have been dramatically overexpressed. This not only perturbs the normal stochiometry of the proteins being examined (that is an important factor here), it leads to cell cycle and morphological defects, with post-translational regulation likely to be disrupted significantly. We thank the reviewer for the suggestion that Cam2 IQ binding may simply inhibit growth (by disrupting lever arm rigidity/structure and motor activity), and we have added this model to the Discussion.

3) It is unclear why the authors chose to use only the non-phosphorylated myo1.S742A allele in all of the in vivo experiments and only the phosphomimetic myo1.S742D allele in all of the in vitro experiments. Both alleles should be used in under both assay conditions to provide essential internal consistency and appropriate controls in all of the experiments.

In vitrothe recombinant protein remains unmodified and therefore we need to mimic phosphorylation, which is why use the phosphomemetic aspartic acid substitution. In yeast cells however, the reverse is true, and the most appropriate method to understand the impact of the phosphorylation event upon the cell, is to replace the serine for a phosphorylation resistant alanine residue. We could see no significant benefit in replacing the serine for an aspartic acid within the yeast cell, and repeating each analysis again for this strain, as it would not provide further insight into the mechanism described here.

4) Figure 3 presents evidence suggesting that the myo1.S742A mutant strain undergoes atypical polar cell growth, but no model or explanation for this phenotype is given. Statistical analysis of the sub-cellular distribution of the mNeonGreen.Myo1 signals in Figure 4D and the distribution of actin may help to explain this observation. With regard to the misregulation of polar growth caused by the myo1.S742A allele, this phenotype does not seem very deleterious and certainly does not significantly affect the growth rate of the cells (Figure 4C). Can this allele be combined with any other genetic backgrounds that are known to affect polar growth to exacerbate this phenotype?

We have done as the reviewer requested and compared polarised growth in *myo1* and *myo1.S742A* cells lacking combinations of *cam2∆*, and the *tea4∆* deletion. Tea4 was chosen for as it demonstrates a strong polarity defect, role in regulating the actin cytoskeleton, and regulating the onset of bipolar cell growth (Figure 5A). Intriguingly, while the *myo1.S742A* allele did not exacerbate *tea4∆* associated defects, the *cam2∆* did. This provides another indication that Cam2 and Myo1 play independent roles during the mitotic cell cycle. Text has been added to describe this result.

5) Figure 2 presumably contains a wealth of useful data, but is extremely hard to interpret due to a lack of detailed experimental information in the corresponding figure legend, as well as a general lack of labels and titles on the graphs (specifically panels B, C, and I).

Figure 2 has now been simplified significantly and contains fewer panels to simplify interpretation for the reader.

[Editors' note: the author responses to the re-review follow.]

Essential revisions:1) The authors show that Myo1^S742^ phosphorylation is dependent on Gad8 and Ste20 (rictor subunit in TORC2). However, whether this is a direct or indirect phosphorylation remains unknown. The data would be more convincing if the authors would replace Figure 1B with a better quality blot, more rigorously quantify the levels of phosphorylated Myo1, and also include a tor1∆ mutant strain to their analysis. In the absence of biochemical experiments to test Myo1 phosphorylation by Gad8, they should clearly state that the data do not necessarily support direct phosphorylation of Myo1 by Gad8.

We have repeated this experiment > 10 times, and the western blot we show in Figure 1B represents the typical optimal western. The average relative phosphorylation shown was calculated from 5 typical equivalent experiments (stated in legend). We have clarified the figure legend and included the standard deviation to the figure to clarify quantification. Similarly, we have now included n, and s.d. values for Figure 1D.

The *ste20∆* specifically abolishes the TORC2 signalling pathway, and therefore we focused on this throughout the study. Strain availability would not allow the *tor1∆* experiment to be undertaken with appropriate rigour within the time allowed. However its absence does not affect the conclusions made within this study.

We have modified the text to indicate Gad8 dependent phosphorylation of Myo1 may not be direct.

2) For the studies on cell cycle regulation of Gad8 and Myo1^S742^-P, the authors should also test if the level of TORC2-Gad8 activity is regulated in a cell cycle dependent manner. If this is not the case, the authors should tone down their conclusions that Gad8 is responsible for the cell cycle phosphorylation of Myo1.

We have deleted and modified the text and toned down the conclusion that Gad8 is responsible for the mitotic and meiotic cell cycle dependent phosphorylation of Myo1.